# High-content screening identifies a small molecule that restores AP-4-dependent protein trafficking in neuronal models of AP-4-associated hereditary spastic paraplegia

Afshin Saffari[1,2], Barbara Brechmann[1], Cedric Böger[1], Wardiya Afshar Saber [1], Hellen Jumo [1,3], Dosh Whye [3], Delaney Wood[3], Lara Wahlster[4], Julian E. Alecu[1], Marvin Ziegler [1], Marlene Scheffold[1], Kellen Winden[1], Jed Hubbs [3], Elizabeth D. Buttermore [3], Lee Barrett[3], Georg H. H. Borner [5], Alexandra K. Davies [5,6], Darius Ebrahimi-Fakhari [1,7,8] ✉ & Mustafa Sahin [1,3,8]

Unbiased phenotypic screens in patient-relevant disease models offer the potential to detect therapeutic targets for rare diseases. In this study, we developed a high-throughput screening assay to identify molecules that correct aberrant protein trafficking in adapter protein complex 4 (AP-4) deficiency, a rare but prototypical form of childhood-onset hereditary spastic paraplegia characterized by mislocalization of the autophagy protein ATG9A. Using high-content microscopy and an automated image analysis pipeline, we screened a diversity library of 28,864 small molecules and identified a lead compound, BCH-HSP-C01, that restored ATG9A pathology in multiple disease models, including patient-derived fibroblasts and induced pluripotent stem cell-derived neurons. We used multiparametric orthogonal strategies and integrated transcriptomic and proteomic approaches to delineate potential mechanisms of action of BCH-HSP-C01. Our results define molecular regulators of intracellular ATG9A trafficking and characterize a lead compound for the treatment of AP-4 deficiency, providing important proof-of-concept data for future studies.

Despite remarkable advances in our ability to delineate the genetic causes of rare neurological diseases, it is estimated that specific therapies exist for less than 5%[1]. Thus, there is a significant unmet need for developing and implementing platforms for drug discovery. Informed by disease-relevant cellular phenotypes, automated and unbiased cell-based high-throughput small molecule screens have the potential to uncover therapeutic targets[2–6].

[1]Department of Neurology & F.M. Kirby Neurobiology Center, Boston Children's Hospital, Harvard Medical School, Boston, MA 02115, USA. [2]Division of Child Neurology and Inherited Metabolic Diseases, Heidelberg University Hospital, Heidelberg, Germany. [3]Rosamund Stone Zander Translational Neuroscience Center, Boston Children's Hospital, Harvard Medical School, Boston, MA 02115, USA. [4]Department of Hematology & Oncology, Boston Children's Hospital & Dana-Farber Cancer Institute, Harvard Medical School, Boston, MA 02115, USA. [5]Department of Proteomics and Signal Transduction, Max-Planck-Institute of Biochemistry, Martinsried 82152, Germany. [6]School of Biological Sciences, Faculty of Biology, Medicine and Health, Manchester Academic Health Science Centre, University of Manchester, Manchester M13 9PT, UK. [7]Movement Disorders Program, Department of Neurology, Boston Children's Hospital, Harvard Medical School, Boston, MA 02115, USA. [8]These authors jointly supervised this work: Darius Ebrahimi-Fakhari, Mustafa Sahin. ✉e-mail: darius.ebrahimi-fakhari@childrens.harvard.edu

Adapter protein complex 4 (AP-4)-related hereditary spastic paraplegia (AP-4-HSP), which comprises *AP4B1*-associated SPG47 (OMIM #614066), *AP4M1*-associated SPG50 (OMIM #612936), *AP4E1*-associated SPG51 (OMIM #613744) and *AP4S1*-associated SPG52 (OMIM #614067), is a rare but prototypical form of childhood-onset complex hereditary spastic paraplegia (HSP) and an important genetic mimic of cerebral palsy[7,8]. Children with AP-4-HSP present with features of both a neurodevelopmental disorder (e.g., early-onset global developmental delay and seizures, microcephaly, and developmental brain malformations) and a neurodegenerative disease (e.g., progressive spasticity and weakness, loss of ambulation, and extrapyramidal movement disorders)[7–10]. AP-4-HSP is caused by bi-allelic loss-of-function variants in any of the four AP-4 subunits (ε, β4, μ4, σ4), leading to impaired AP-4 assembly and function[11–15]. AP-4 is an obligate heterotetrameric protein complex[16–18] that mediates transport from the trans-Golgi network (TGN) to the cell periphery, including sites of autophagosome biogenesis[19,20]. Three independent groups identified the core autophagy protein and lipid scramblase ATG9A as a major cargo of AP-4[11–14,21], linking loss of AP-4 function to defective autophagy[22,23]. AP-4 deficiency in non-neuronal[11,12,21,24,25] and neuronal cells[13–15] leads to an accumulation of ATG9A in the TGN, including in iPSC-derived neurons from AP-4-HSP patients[15]. From this body of work and overlapping neuronal phenotypes of AP-4[13,14,26,27] and *Atg9a*[28] knockout mice, the following working model for AP-4 deficiency emerges: (1) AP-4 is required for trafficking of ATG9A from the TGN; (2) loss-of-function variants in AP-4 subunits lead to a loss of AP-4 function; (3) ATG9A accumulates in the TGN leading to a reduction of axonal delivery of ATG9A; (4) lack of ATG9A at the distal axon impairs autophagy leading to axonal degeneration. Other AP-4 cargo proteins identified to date include the poorly characterized transmembrane proteins SERINC1 and SERINC3[12] and the endocannabinoid-producing enzyme DAG lipase beta (DAGLB)[29].

In this study, we leverage intracellular ATG9A mislocalization as a cellular readout for AP-4 deficiency to develop a large-scale, automated, multiparametric, unbiased phenotypic small molecule screen for modulators of ATG9A trafficking in patient-derived cellular models. We employed this platform to screen a diversity library of 28,864 small molecules in AP-4-deficient patient fibroblasts and identified 503 compounds that re-distribute ATG9A from the TGN to the cytoplasm. Through a series of orthogonal assays in neuronal cells, including differentiated *AP4B1^KO* SH-SY5Y cells and human induced pluripotent stem cell (hiPSC)-derived neurons from AP-4-HSP patients, we defined a series of 5 compounds that restore neuronal phenotypes of AP-4-deficiency. In a comprehensive multiparametric analysis, a small molecule, termed BCH-HSP-C01, emerged as a lead compound with an EC50 of ~5 μM. Target deconvolution strategies using transcriptomic and proteomic profiling revealed that BCH-HSP-C01 modulates intracellular vesicle trafficking and increases autophagic flux, potentially through differential expression of several RAB (Ras-associated binding) proteins.

Our findings demonstrate the ability of carefully designed high-throughput screens to identify potential molecular mechanisms involved in AP-4 deficiency and support the development of BCH-HSP-C01 as a therapeutic for AP-4-HSP.

## Results

### Primary screening of 28,864 compounds in fibroblasts from AP-4-HSP patients identifies 503 active compounds

A diversity library of 28,864 small molecules was provided by Astellas Pharma Inc. Compounds were arrayed to single wells in 384-well microplates, and one well per compound was screened. The primary screen was conducted in fibroblasts from a well-characterized patient with core clinical features of SPG47[8,21] and bi-allelic loss-of-function variants in *AP4B1* (NM_001253852.3: c.1160_1161del (p.Thr387ArgfsTer30) / c.1345A>T (p.Arg449Ter)) (Fig. 1a, b). Fibroblasts from the sex-matched parent (unaffected heterozygous carrier) served as controls. The assay was fully automated, miniaturized to 384-well microplates, and compounds were added for 24 h at a single concentration of 10 μM (Fig. 1c).

The ATG9A ratio (ATG9A fluorescence intensity inside the TGN vs. in the cytoplasm) was used as the primary assay metric, as established previously[15,21]. The population distributions of the subcellular ATG9A signal inside and outside the TGN, at the level of single cells for negative (bi-allelic loss-of-function, LoF/LoF) and positive (heterozygous carriers, WT/LoF) controls are shown in Fig. 1d, e. ATG9A ratios demonstrated symmetrical and approximately normal distributions and robust separation of both groups (Fig. 1f). Cell counts were similar for positive and negative controls, excluding cell death or changes in proliferation rates as possible confounding factors (Fig. 1g). To test for reproducibility across replicates, assay plates were randomly sampled into two sets, and similar positions on the assay plates were plotted against each other (Fig. 1h, i). Random sampling was simulated 100 times, and mean correlation coefficients were calculated. Using the ATG9A ratio (Fig. 1i) as a primary readout resulted in higher replicate correlation (mean $r = 0.90 \pm 0.002$ SD) compared to absolute ATG9A intensities (Fig. 1h) (mean $r = 0.82 \pm 0.0008$ SD). ATG9A ratios showed robust discriminative power between positive and negative controls (LoF/LoF mean: $1.34 \pm 0.05$ SD, $n = 1312$ wells vs. WT/LoF mean: $1.1 \pm 0.02$ SD, $n = 1312$ wells, Mann-Whitney U test, $p < 0.0001$) (Fig. 1j). The ATG9A ratio as the primary outcome metric was further supported by a generalized linear model, which demonstrated high specificity and sensitivity (Fig. 1k, AUC: 0.96). Source data for assay performance are provided in Source Data file 1.

Throughout the screen, assay performance was monitored using established quality control metrics for cell-based screens (Z' robust ≥0.3, strictly standardized median difference ≥3, and an inter-assay coefficient of variation ≤10%)[30–32]. All assay metrics were calculated for positive and negative controls of the same assay plate to avoid bias by inter-plate variability. Predefined thresholds were met by all assay plates (Supplementary Fig. 1a and Source Data file 2). The results of the primary screen are summarized in Fig. 1l, m, and the complete dataset is provided in Source Data file 3.

Of the 28,864 compounds, 26 were excluded due to non-quantifiable ATG9A signal, exceptionally low cell counts or imaging artifacts. The remaining 28,838 compounds were evaluated for changes in cell count and the ATG9A ratio. The vast majority ($n = 26,961$, 93.5%) did not show any significant reduction in the ATG9A ratio (defined as a reduction by at least 3 SD). 1,435 (5.0%) compounds were excluded due to toxicity, defined as a reduction in the mean cell count by at least 2 SD compared to the negative controls. Only a small subset of 503 compounds (1.7%) reduced the ATG9A ratio by 3 or more SD compared to negative controls (Fig. 1m). Of these, 61 (0.2%) also reduced cell counts, while the remaining 442 (1.5%) showed no toxicity.

In summary, from this high-throughput primary screen, 503 active compounds were identified and selected for further testing.

### Counter-screen in fibroblasts from AP-4-HSP patients confirms 16 compounds that lead to a dose-dependent redistribution of ATG9A

To validate the 503 active compounds identified in the primary screen, compounds were retested for dose-dependency using an 11-point dose range (range: 40 nM to 40 μM) (Fig. 2a). Source data for the secondary screen are provided in Source Data file 4. All concentrations were screened in biological duplicates and subjected to the same quality control metrics as in the primary screen (Supplementary Fig. 1b and Source Data file 5). Similar to the results from the primary screen, ATG9A ratios for negative and positive controls showed a robust separation (LoF/LoF mean: $1.4 \pm 0.07$ (SD), vs. $n = 269$ wells vs. WT/LoF mean: $1.12 \pm 0.02$ (SD), $n = 269$ wells, Mann-Whitney U test, $p < 0.0001$, Fig. 2b). Activity in the secondary screen was defined as the ability to reduce the

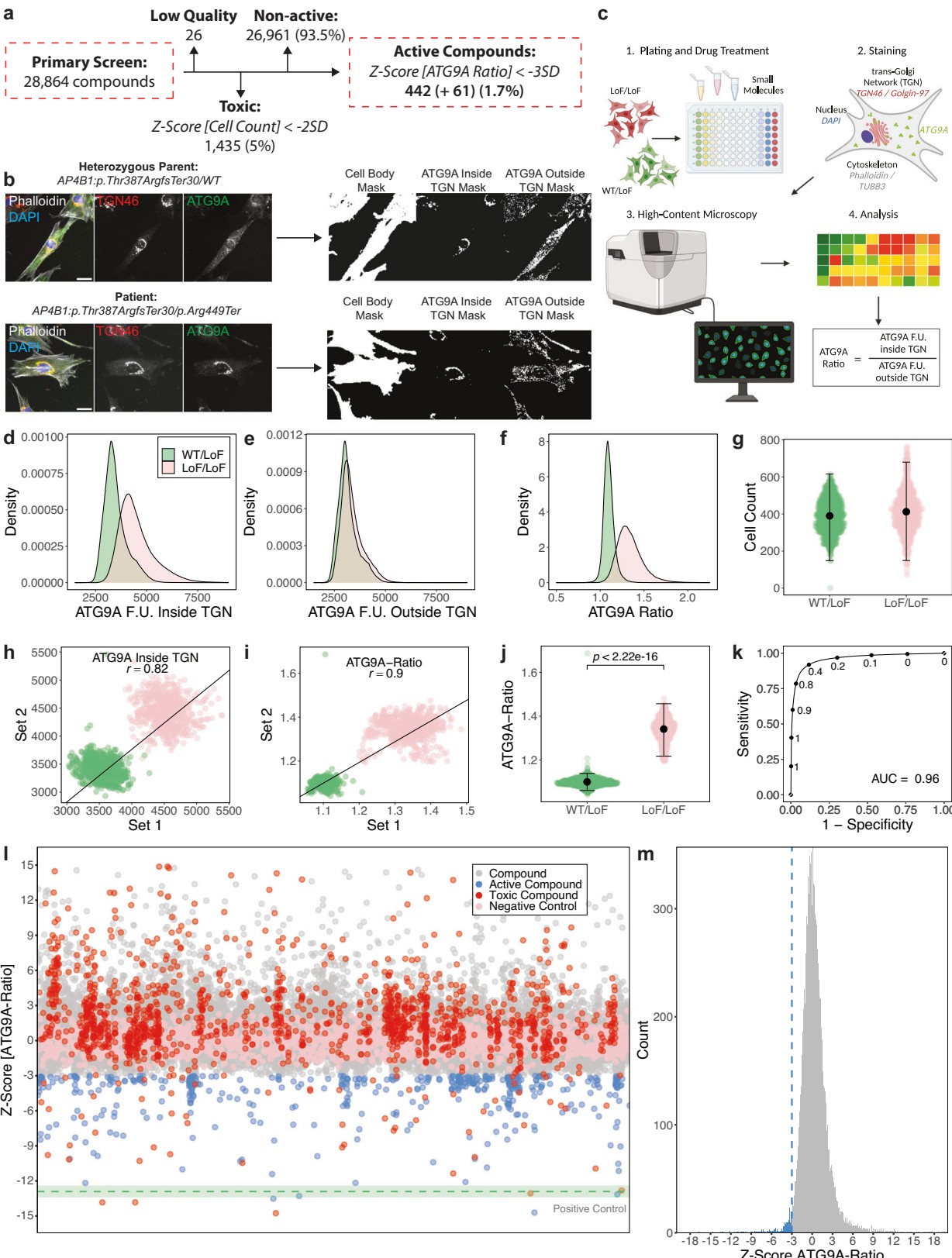

ATG9A ratio by at least 3 SD in both replicates and at least 2 different concentrations, without exerting toxicity. 51 compounds (10.1%) met these a priori defined criteria (Supplementary Fig. 2a, b). After manually verifying image quality and validating dose-response relationships, compounds were triaged (Fig. 2a and Supplementary Fig. 2a, b). Seventeen compounds demonstrated a clear and reproducible dose-

response relationship without evidence of image artifacts or auto-fluorescence. The EC50 for most compounds was in the low micromolar range (median: 4.66 μM, IQR: 8.63, Fig. 2). 34 compounds were found to carry autofluorescence or imaging artifacts and were thus excluded from further testing (Supplementary Fig. 2b). One active compound was unavailable from the manufacturer and was removed.

**Fig. 1 | Establishment of a cell-based phenotypic small molecule screening platform using ATG9A translocation as a surrogate for AP-4 function and primary screening of 28,864 small molecule compounds. a** Overview of the primary screen of 28,864 small molecules in fibroblasts from a patient with bi-allelic LoF variants in *AP4B1*. **b** Illustration of the automated image analysis pipeline. Representative images of patient fibroblasts (negative control, LoF/LoF) and their sex-matched heterozygous parent (positive control, WT/LoF) are shown. Scale bar: 20 μm. **c** Overview of the high-throughput platform. Created with BioRender.com. **d–f** Distribution of ATG9A fluorescence intensities inside (**d**) and outside (**e**) the TGN, as well as ATG9A ratios (**f**) on a per cell basis ($n_{WT/LoF}$ = 99,927, $n_{LoF/LoF}$ = 119,522). **g** Cell counts as per well means of 1312 wells per condition from 82 independent plates. Means are shown as black dots; whiskers represent ±1.5 x IQR. **h, i** Replicate plots were generated by random sampling of the 82 plates from the primary screen in two groups. Similar positions on the assay plates were plotted against each other with respect to ATG9A fluorescence intensities inside the TGN (**h**) and ATG9A ratios (**i**). Replicate correlations were assessed by averaging the Pearson correlation coefficients ($r$) of 100 random sampling tests. **j** Discriminative power of the ATG9A ratio in separating positive and negative controls. Statistical testing was done using the Mann-Whitney U test. *P*-values are two-sided. Data points represent per well means of 1312 wells per condition from 82 independent plates. Means are shown as black dots; whiskers represent ±1.5 x IQR. **k** To test the robustness of separation of the ATG9A ratio between positive and negative controls, a dataset containing measurement for 99,927 WT/LoF and 119,522 LoF/LoF cells was partitioned into a training set (70% of data) and a test set (30%). The performance of a generalized linear model is shown in (**k**). The AUC is 0.96. **l** Impact of 28,864 compounds applied for 24 h at a concentration of 10 μM. *Z*-scores for the ATG9A ratio are shown. All data points represent per well means. The mean of the positive control is shown as a green dotted line. The green shaded areas represent ± 1 SD. **m** Distribution of *Z*-scores of all non-toxic 27,403 compounds. Active compounds are highlighted in blue.

In summary, a counter-screen in AP-4-deficient patient fibroblasts confirmed and established dose-dependent effects on intracellular ATG9A distribution for 16 compounds (Fig. 2c).

## Orthogonal assays in neuronal models of AP-4-deficiency confirm 5 active compounds

To validate active compounds from the secondary screen in a human cell line with neuron-like properties, the ATG9A assay was optimized for neuroblastoma-derived SH-SY5Y cells following a 5-day neuronal differentiation protocol with retinoic acid[33] (Fig. 3a). SH-SY5Y cells with stable expression of *AP4B1*-targeting CRISPR/Cas9 machinery (*AP4B1^KO*)[12] served as negative controls while *AP4B1*-wildtype (*AP4B1^WT*) cells were used as positive controls. All 16 active compounds were tested in an 8-point dose range (50 nM to 30 μM) with a treatment duration of 24 h. Quantification of the ATG9A ratio in differentiated SH-SY5Y cells showed a robust separation between control conditions (*AP4B1^KO*: 1.80 ± 0.06 (SD), $n$ = 158 wells vs. *AP4B1^WT*: 1.17 ± 0.03 (SD), $n$ = 160 wells, Mann-Whitney U test, $p$ < 0.0001, Fig. 3b, Source Data file 6). Compounds were evaluated based on their dose-dependent reduction of the ATG9A ratio and absence of cell toxicity. Eleven of 16 compounds were excluded due to lacking activity ($n$ = 7), suspicion of artifacts or autofluorescence ($n$ = 3), or obvious changes in cellular morphology ($n$ = 1) (Supplementary Fig. 3). Of the five remaining compounds, three restored the ATG9A ratio to levels of wildtype controls (BCH-HSP-F01, BCH-HSP-G01 and BCH-HSP-H01) while two compounds (BCH-HSP-B01 and BCH-HSP-C01) led to a reduction by at least 3 SD at higher concentrations (Fig. 3c–h). The chemical structures and properties of these five compounds are summarized in Supplementary Fig. 4.

To assess whether these effects were specific to ATG9A or similar effects were also present for other AP-4 cargo proteins, we turned to a second neuronal AP-4 cargo protein, DAGLB[29]. Similar to ATG9A, the DAGLB ratio (DAGLB fluorescence intensity in the TGN vs. in the cytoplasm) showed a robust separation between *AP4B1^WT* and *AP4B1^KO* cells (*AP4B1^KO*: 1.80 ± 0.1 (SD), $n$ = 192 wells vs. *AP4B1^WT*: 1.36 ± 0.07 (SD), $n$ = 192 wells, Mann-Whitney U test, $p$ < 0.0001, Fig. 3i, Source Data file 6). All active compounds, except for BCH-HSP-B01, showed activity in the DAGLB assay, suggesting a broader effect on the trafficking of at least 2 AP-4 cargo proteins from the TGN (Fig. 3j–o). Again, BCH-HSP-F01, BCH-HSP-G01 and BCH-HSP-H01 (Fig. 3l–n) resulted in normalization of the intracellular DAGLB distribution, while BCH-HSP-C01 led to a moderate reduction of DAGLB ratios at higher concentrations (Fig. 3k).

Since small molecules can have pleiotropic effects on cellular functions and organellar morphology, we adopted a multiparametric morphological profiling approach[34]. Eighty-five measurements of the nucleus, cytoskeleton, global cell morphology, the TGN, and ATG9A vesicles were automatically computed for each image, serving as a rich and unbiased source for interrogating biological perturbations induced by compound treatment (Source Data file 6). Principal component analysis was used to reduce dimensionality and cluster images based on their properties (Fig. 4a, b and Supplementary Fig. 5). Positive and negative controls clustered closely together and were separated only by the ATG9A signal (Fig. 4b and Supplementary Fig. 5a). BCH-HSP-C01 showed properties comparable to positive and negative controls, suggesting little off-target effects (Fig. 4b and Supplementary Fig. 5c). BCH-HSP-B01, BCH-HSP-F01, BCH-HSP-G01 and BCH-HSP-H01, however, changed cellular morphology in a dose-dependent manner (Fig. 4b and Supplementary Fig. 5b, d–f), with changes mainly driven by the first principal component, accounting for 31.1% of the observed variance (Fig. 4c). To decipher the phenotypic alterations responsible for these changes, the Pearson correlation coefficients of the first principal component with each measurement were calculated (Fig. 4d). Features with a correlation coefficient >0.75 were selected to define morphological profiles (Fig. 4e). Interestingly, TGN fluorescence intensity and morphology seemed to be the most significant drivers for the separation, suggesting that disruption of TGN integrity potentially biased the assessment of ATG9A ratios in cells treated with compounds BCH-HSP-B01, BCH-HSP-F01, BCH-HSP-G01 and BCH-HSP-H01 (Fig. 4b and Supplementary Fig. 5b, d–f).

Following these analyses, TGN fluorescence intensity and morphological measures such as TGN area and elongation, as well as compactness and roughness, as indicators of the complexity of the TGN, were quantified for cells treated with all five active compounds (Fig. 4f, g). While BCH-HSP-C01 showed stable TGN signal and morphology across all assessed measurements, the other compounds induced some degree of change in a dose-dependent manner (Fig. 4f, g). Of note, these changes to TGN morphology were not detectable by visual inspection but only delineated through an automated analysis of ~600 images containing ~30,000 cells per group, showcasing the power of our automated, unbiased, high-throughput platform.

## BCH-HSP-C01 restores ATG9A and DAGLB trafficking in hiPSC-derived neurons from AP-4-HSP patients

Informed by the findings in differentiated *AP4B1^KO* SH-SY5Y cells, we next investigated whether these results would translate to human neurons. hiPSCs from patients with AP-4-HSP due to bi-allelic loss-of-function variants in *AP4M1* (NM_004722.4: c.916C>T (p.Arg306Ter) / c.694dupG (p.Glu232GlyfsTer21)) and *AP4B1* (NM_001253852.3: c.1160_1161del (p.Thr387ArgfsTer30) / c.1345A>T (p.Arg449Ter)) were generated[35,36] and differentiated into glutamatergic cortical neurons using established protocols[15,37,38]. hiPSC-derived neurons from sex-matched parents (unaffected heterozygous carriers) served as controls (Fig. 5a and Source Data file 7). Baseline quantification of ATG9A ratios in DIV (day in vitro) 14 neurons treated with vehicle for 24 h showed robust separation between patient and control lines, exceeding the differences observed in AP-4-deficient fibroblasts and differentiated SH-SY5Y cells (SPG50 patient mean: 4.31 ± 0.4 (SD), $n$ = 60

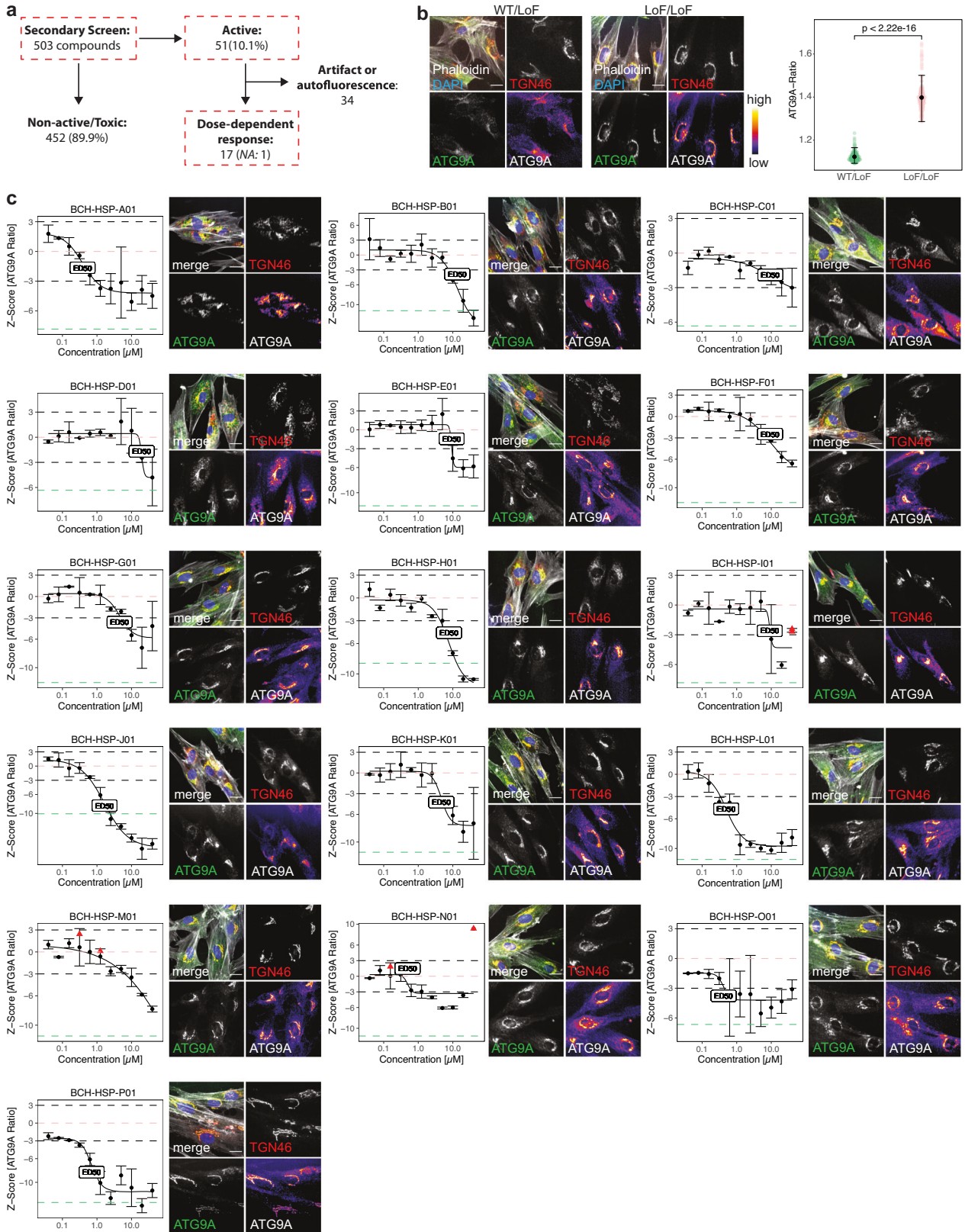

wells vs. heterozygous control: $1.56 \pm 0.12$ (SD), $n = 60$ wells, Mann-Whitney U test, $p < 0.0001$, Fig. 5b). Neurons were treated for 24 h in 8-point dose titration experiments. BCH-HSP-B01 and BCH-HSP-G01 lacked activity on the ATG9A ratio and were thus excluded (Fig. 5d). BCH-HSP-C01, BCH-HSP-F01 and BCH-HSP-H01, by contrast, showed a robust reduction in the ATG9A ratio (Fig. 5e, f; Supplementary Fig. 6). A

multiparametric analysis showed that, similar to observations in $AP4B1^{KO}$ SH-SY5Y cells, only BCH-HSP-C01 preserved TGN integrity (Fig. 5f), while BCH-HSP-F01 and BCH-HSP-H01 impacted TGN morphology, suggesting off-target effects (Fig. 5e). Based on its favorable profile, BCH-HSP-C01 was selected as a lead compound and was re-synthesized for further testing.

**Fig. 2 | Counter-screen in fibroblasts from AP-4-HSP patients confirms 16 compounds that lead to dose-dependent redistribution of ATG9A. a** Overview of the counter-screen of the 503 active compounds identified in the primary screen. To assess for dose-dependent effects, compounds were screened in AP-4-HSP patient-derived fibroblasts in 384-well microplates using 11-point titrations ranging from 40 nM to 40 μM. All concentrations were screened in duplicates. Active compounds were a priori defined as those reducing the ATG9A ratio by at least 3 SD compared to negative controls in more than one concentration. Toxicity was defined as a reduction of the cell count of at least 2 SD compared to negative controls. **b** Baseline differences in the ATG9A distribution in WT/LoF ($n = 269$) vs. LoF/LoF ($n = 269$) fibroblasts. Data points represent per well means of 269 wells per condition from 17 independent plates. Means are shown as black dots; whiskers represent ±1.5 x IQR. Statistical testing was done using the Mann-Whitney U test. *P*-values are two-sided. **c** Dose-response curves were fitted using a four-parameter logistic regression model, and EC50 concentrations were calculated. All concentrations were tested in biologic duplicates. Black dots and error bars represent mean ± 1 SD. Black dashed lines represent the a priori-defined thresholds of ± 3 SD compared to the negative control (LoF/LoF). Red triangles represent toxic concentrations based on the a priori-defined threshold of a reduction of cell counts of at least 2 SD compared to the negative control. The salmon-colored dashed line represents the mean of negative controls, while the green-colored dashed line depicts the mean of the positive controls (WT/LoF). Representative images of the EC50 are shown for each active compound. Representative images show a merge of the 4 channels: Phalloidin (gray), DAPI (blue), TGN46 (red) and ATG9A (green), as well as the TGN46 and ATG9A channels in greyscale. For a better illustration of differences in ATG9A signals, the fluorescence intensities of the ATG9A channel are additionally shown using a color lookup table. Scale bar: 20 μm. NA: not available.

To investigate the time- and dose-dependent effect of BCH-HSP-C01, we used *AP4B1[KO]* SH-SY5Y cells to conduct time-series experiments with different concentrations of BCH-HSP-C01 (Fig. 5g, h, Source Data file 6). All concentrations tested show a maximal effect on ATG9A translocation after 72–96 h of treatment (Fig. 5g, h, Source Data file 6), exceeding the effects seen after 24 h (Fig. 3d).

Turning back to hiPSC-derived neurons, prolonged treatment of BCH-HSP-C01 for 72 h to test for ATG9A and DAGLB translocation demonstrated that BCH-HSP-C01 was able to restore ratios of both AP-4 cargo proteins to levels close to controls with an EC50 of ~5 μM, while maintaining a favorable profile (Fig. 5i, Source Data file 7). This greater effect on ATG9A distribution, compared to the ~50% reduction of the ATG9A ratio at 24 h treatment, again suggests a time- and dose-dependent effect. BCH-HSP-C01 changed the ATG9A ratio by simultaneously decreasing ATG9A intensities inside the TGN and increasing cytoplasmic ATG9A levels, suggesting ATG9A translocation as the most likely mechanism of action. No changes in TGN morphology or any other cellular measurements were observed, indicating overall preservation of cellular morphology and little off-target effects. A similar pattern was observed with respect to DAGLB translocation (Fig. 5i). These findings were confirmed in a second set of experiments in hiPSC-derived neurons from a patient with SPG47 (Fig. 5j, Source Data file 7), demonstrating that the findings extend to other forms of AP-4-deficiency.

Prior work in neurons isolated from AP-4-deficient mice[13,14] highlighted the depletion of axonal ATG9A pools. In hiPSC-derived neurons from individuals with SPG50 and SPG47, we observed a reduction of ATG9A puncta density in neurites. BCH-HSP-C01 treatment for both 24 h and 72 h restored neurite ATG9A puncta density to levels similar to controls (Fig. 5k–m, Source Data file 7).

Taken together, BCH-HSP-C01 emerged as a robust modulator of ATG9A and DAGLB trafficking in human neurons from patients with AP-4 deficiency.

## Target deconvolution using transcriptomic and proteomic analyses delineates putative mechanisms of action for BCH-HSP-C01

To explore potential mechanisms of action of BCH-HSP-C01 in an unbiased manner, we used a multi-omics approach, combining bulk RNA sequencing and unbiased label-free quantitative proteomics (source data are provided in Source Data files 8–10).

First, RNA sequencing was conducted in differentiated *AP4B1[WT]* and *AP4B1[KO]* SH-SY5Y cells treated for 72 h with either vehicle or compound BCH-HSP-C01 (5 μM, Source Data file 8). Analysis of differential gene expression identified few significant transcriptional changes in response to BCH-HSP-C01 treatment, suggesting that this compound does not elicit major alterations in gene expression or induce many off-target effects (Supplementary Fig. 7). Since changes in gene expression caused by short-duration small molecule treatments might not reach predefined cutoffs for standard differential

expression analyses, and because compounds might affect groups of genes in shared pathways rather that modifying single target genes, we adapted an unbiased and unsupervised network approach to identify groups of co-expressed genes. Hierarchical clustering of samples showed that treatment with BCH-HSP-C01, regardless of cell line, was the main differentiator in our dataset (Fig. 6a). To identify the gene networks responsible for these changes, weighted gene co-expression network analysis (WGCNA)[39,40] was used to group the 18,506 expressed genes into 36 co-expression modules (Fig. 6b). Gene expression profiles within each module were summarized using the "module eigengene" (ME), defined as the first principal component (PC) of a module[41]. Within each module, the association of MEs with measured traits was examined by correlation analysis (Fig. 6c). Eight modules that showed an absolute correlation coefficient >0.5 were selected for further evaluation. For these selected modules, ME-based connectivity was determined for every gene by calculating the absolute value of the Pearson correlation between the expression of the gene and the respective ME, producing a quantitative measure of module membership (MM). Similarly, the correlation of individual genes with BCH-HSP-C01 treatment was computed, defining gene significance (GS) for BCH-HSP-C01. Using the GS and MM, an intramodular analysis was performed, allowing identification of genes that have a high correlation with treatment as well as high connectivity to their modules (Fig. 6d). Five modules were significantly related to BCH-HSP-C01 treatment, defined as showing an absolute correlation coefficient between MM and GS >0.5 (Fig. 6e). A list of the genes contained in each module along with their module membership is provided in Source Data file 9. To summarize the biological information contained in these modules of interest, gene ontology (GO) analysis was performed, which demonstrated enrichment in biological pathways in three out of the five assessed modules (Fig. 6f). The 'blue module' showed downregulation of pathways involved in axonogenesis, actin filament organization and proteasome-mediated pathways. The 'light-yellow module' contained genes involved in ER stress response, amino acid metabolism and transcription. Finally, the 'mediumpurple3 module' depicted the upregulation of genes involved in vesicular transport, particularly involving TGN and ER-associated transport, as well as membrane and vesicle dynamics. This last module showed the highest gene ratios (defined as the percentage of total differentially expressed genes in the given GO term) and lowest *P*-values of all differentially regulated pathways across all modules, suggesting the upregulation of alternative vesicle-mediated transport mechanisms by compound BCH-HSP-C01 (Fig. 6f).

To assess whether similar themes would emerge on the protein level, we next used unbiased quantitative proteomics in both differentiated SH-SY5Y cells (*AP4B1[KO]* and *AP4B1[WT]*) and hiPSC-derived neurons (patient with *AP4B1*-associated SPG47 and control) treated for 72 h with either vehicle or compound BCH-HSP-C01 (5 μM). After quality filtering, 8,141 unique proteins in SH-SY5Y cells and 7386 unique proteins in hiPSC-derived neurons were quantified. Differential

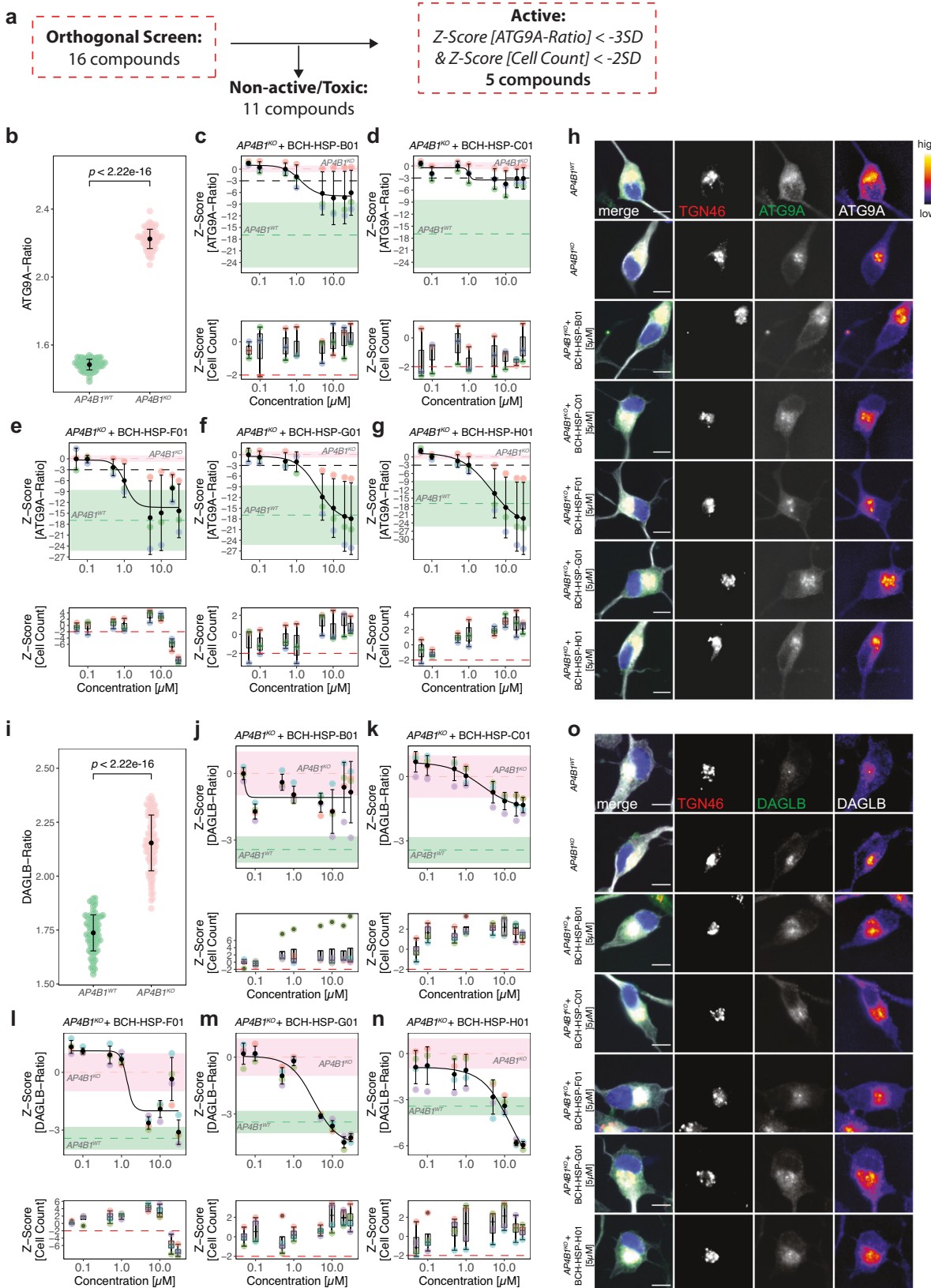

enrichment analyses for both cell lines are shown in Fig. 7a, b, and source data are provided in Source Data file 10. As expected, baseline quantification of differentially expressed proteins in *AP4B1^KO* SH-SY5Y cells showed downregulation of AP-4 subunits, AP4B1, AP4E1 and AP4M1, and increased ATG9A and DAGLB levels, as reported in other models of AP-4 deficiency[11-13,29] (Supplementary Fig. 8a). Moreover,

additional dysregulation of proteins involved in autophagy, Golgi dynamics and vesicular transport was identified (Supplementary Fig. 8a, m, Source Data file 10). Of note, we observed upregulation of ATG2A, which has recently been shown to form a complex with ATG9A that facilitates lipid transfer from the endoplasmic reticulum (ER) to the growing phagophore membrane[42-44]. This further supports that

**Fig. 3 | Orthogonal assays in *AP4B1^KO* SH-SY5Y cells confirm 5 active compounds. a** Overview of the orthogonal screen of 16 active compounds in differentiated *AP4B1^KO* SH-SY5Y cells. **b** Baseline differences in ATG9A ratios of *AP4B1^WT* vs. *AP4B1^KO* SH-SY5Y cells were quantified from 160 *AB4B1^WT* and 158 *AB4B1^KO* wells from 5 assay plates. Means are shown as black dots; whiskers represent ±1.5 x IQR. Statistical testing was performed using the Mann-Whitney U test. *P*-values are two-sided. **c–g** Dose-response curves for ATG9A ratios in *AB4B1^KO* cells treated with different compounds. Data points represent per well means from 3 different assay plates. Black dots and error bars represent mean ± 1 SD. Dashed lines show mean *Z*-scores for positive (green) and negative (salmon) controls. Shaded areas represent ± 1 SD. **h** Representative images of the intracellular ATG9A distribution for individual compounds. The merged image shows beta-3 tubulin (gray), DAPI (blue), the TGN46 (red) and ATG9A (green). The TGN46 and ATG9A channels are further

separately depicted in greyscale. Scale bar: 10 μm. **i** Baseline differences of DAGLB ratios in *AP4B1^WT* vs. *AP4B1^KO* cells were quantified from 192 *AB4B1^WT* and 192 *AB4B1^KO* wells from 4 assay plates. Means are shown as black dots; whiskers represent ±1.5 x IQR. Statistical testing was done using the Mann-Whitney U test. *P*-values are two-sided. **j–n** Dose-response curves for DAGLB ratios in *AB4B1^KO* cells treated with different compounds. All data points represent per well means from 4 different assay plates. Black dots and error bars represent mean ± 1 SD. Dashed lines show mean *Z*-scores for positive (green) and negative (salmon) controls. Shaded areas represent ± 1 SD. **o** Representative images of the intracellular DAGLB distribution for individual compounds. The merge shows beta-3 tubulin (gray), DAPI (blue), the TGN46 (red) and DAGLB (green). The TGN46 and DAGLB channels are further separately depicted in greyscale. Scale bar: 10 μm.

autophagosome biogenesis is dysregulated in AP-4-deficient cells. PCA analysis of SH-SY5Y cells demonstrated 4 distinct clusters separated by BCH-HSP-C01 treatment (PC1, explaining 12.3% of variance) and genotype (PC2, explaining 8.7% of variance) (Fig. 7a). Testing of vehicle vs. BCH-HSP-C01 treated cells showed broadly similar groups of dysregulated proteins in *AP4B1^WT* and *AP4B1^KO* SH-SY5Y cells (Supplementary Fig. 8b–d), suggesting a conserved mechanism of action independent of genotype, which allowed the pooling of cell lines to increase the power of the analysis (Fig. 7a). Similar observations were made for hiPSC-derived neurons (Fig. 7b and Supplementary Fig. 8e–h). Here, cell lines were a stronger discriminator, likely due to heterogeneity of the positive and negative controls, as expected in cell lines derived from different individuals (Supplementary Fig. 8e, n, Source Data file 10). Again, differentially enriched proteins following BCH-HSP-C01 treatment in hiPSC neurons showed a high degree of similarity between patient and control lines (Supplementary Fig. 8f–h), allowing a combined analysis (Fig. 7b).

Despite the heterogeneity in the neuronal samples, significant overlap was observed between the differentially enriched proteins in SH-SY5Y cells and hiPSC-derived neurons. Data sets were thus integrated for a combined analysis, which detected several proteins that were dysregulated across all cell types and genotypes (Supplementary Fig. 8i–l), providing strong evidence that these changes were related to treatment with BCH-HSP-C01 (Fig. 7c). Consistent with the overall changes in gene expression, pathway enrichment analysis using the Reactome database[45] highlighted engagement of intracellular trafficking pathways as a potential mechanism of action for BCH-HSP-C01 (Fig. 7c). Specifically, modulation of RAB proteins involved in vesicle transport emerged as a consistent theme across cell types and genotypes, with the strongest evidence for the upregulation of RAB1B and downregulation of RAB3C and RAB12. Notably, while BCH-HSP-C01 led to a significant change in protein levels of all three RAB protein family members in SH-SY5Y cells, only RAB3C and RAB12 reached significance in hiPSC-derived neurons (Fig. 7d). This overall pattern of RAB protein modulation was further supported by upregulation of the RAB protein geranylgeranyltransferase components A1 (CHM) in SH-SY5Y cells and A2 (CHML) in both SH-SY5Y cells and hiPSC-derived neurons. CHM and CHML play a vital role in tethering RAB proteins to intracellular membranes[46,47]. Additionally, upregulation of transferrin receptor protein 1 (TFRC) was observed (Fig. 7c), consistent with prior reports showing that reduction of RAB12 associates with increased protein levels of TFRC[48]. Collectively, these findings suggest a potential role of RAB proteins in regulating vesicle transport in response to BCH-HSP-C01 treatment.

## RAB3C and RAB12 knockout are involved in BCH-HSP-C01-mediated vesicle trafficking and autophagy

RAB3C and RAB12 displayed the strongest and most consistent protein expression changes in both differentiated SH-SY5Y cells and hiPSC-derived neurons following treatment with BCH-HSP-C01 (Fig. 7d) and were therefore selected for further investigation. Correlation analysis

revealed a strong correlation ($r = 0.93$) between the LFQ intensities of these two proteins in both cell types and across different genotypes in response to BCH-HSP-C01 (Fig. 7e).

To assess whether a correlation was also present on the transcriptional level, mRNA levels of *RAB3C* and *RAB12* in response to BCH-HSP-C01 treatment were analyzed in *AP4B1^WT* and *AP4B1^KO* SH-SY5Y cells. While there was a trend toward a reduction of *RAB3C* and elevation of *RAB12* mRNA levels and correlation analysis demonstrated a moderate inverse correlation, none of these changes reached statistical significance (Supplementary Fig. 9). These findings suggest that RAB3C and RAB12 levels are altered through a post-transcriptional mechanism following treatment with BCH-HSP-C01.

To investigate the potential impact of RAB3C and RAB12 on ATG9A translocation in the AP-4-deficient background, we used CRISPR/Cas9-mediated knockouts of RAB3C and RAB12 in *AP4B1^KO* SH-SY5Y cells (Fig. 8a, b, Supplementary Fig. 10 and Source Data file 11). We found that knockout of RAB12 did not affect ATG9A translocation, while knockout of RAB3C caused a moderate reduction in the ATG9A ratio (Fig. 8a). Combined knockout of RAB3C and RAB12 in *AP4B1^KO* SH-SY5Y cells did not show an additive effect. Interestingly, however, the effects of BCH-HSP-C01 on ATG9A translocation were significantly enhanced by knockout of RAB3C, but not RAB12 alone. Combined knockout of both genes further augmented the effect of BCH-HSP-C01. These findings suggest that both RAB3C alone or in combination with RAB12 play a role in BCH-HSP-C01-mediated ATG9A redistribution.

A converging theme of ATG9A translocation and alteration of RAB protein expression is autophagy. RAB proteins are known modulators of autophagy with key functions in various steps of the pathway[49,50]. ATG9A, a core autophagy protein, acts as a lipid scramblase and promotes autophagosome formation and elongation[43,51–53]. To investigate whether BCH-HSP-C01 leads to changes in autophagic flux, *AP4B1^WT* and *AP4B1^KO* SH-SY5Y cells were treated with BCH-HSP-C01 for 72 h and LC3-I to LC3-II conversion was measured by western blotting (Fig. 8c–f and Supplementary Fig. 10a). Levels of LC3-II were significantly elevated in all cell lines treated with BCH-HSP-C01, suggesting modulation of the autophagy pathway. Co-treatment with bafilomycin A1, which blocks autophagosome-lysosome fusion, led to further LC3-II accumulation, indicating that BCH-HSP-C01 increases autophagic flux (Fig. 8c–f). Blocking the late stages of the autophagy pathway with either bafilomycin A1 or chloroquine reversed the effect of BCH-HSP-C01 on ATG9A translocation in a dose-dependent manner, suggesting that this process requires intact autophagic flux (Fig. 8g–i and Source Data file 12).

Next, since our data suggested a contribution of RAB3C and RAB12 to the effect of BCH-HSP-C01, we investigated the impact of RAB3C and RAB12 knockout in *AP4B1^KO* SH-SY5Y cells with and without BCH-HSP-C01 treatment (Fig. 8j–l and Supplementary Fig. 10b–d). Neither RAB3C nor RAB12 knockout alone led to major changes in baseline or BCH-HSP-C01-enhanced autophagic flux (Fig. 8j, k). However, combined knockout of RAB3C and RAB12, without BCH-HSP-C01 treatment, significantly increased the ratio of LC3-II to LC3-I by

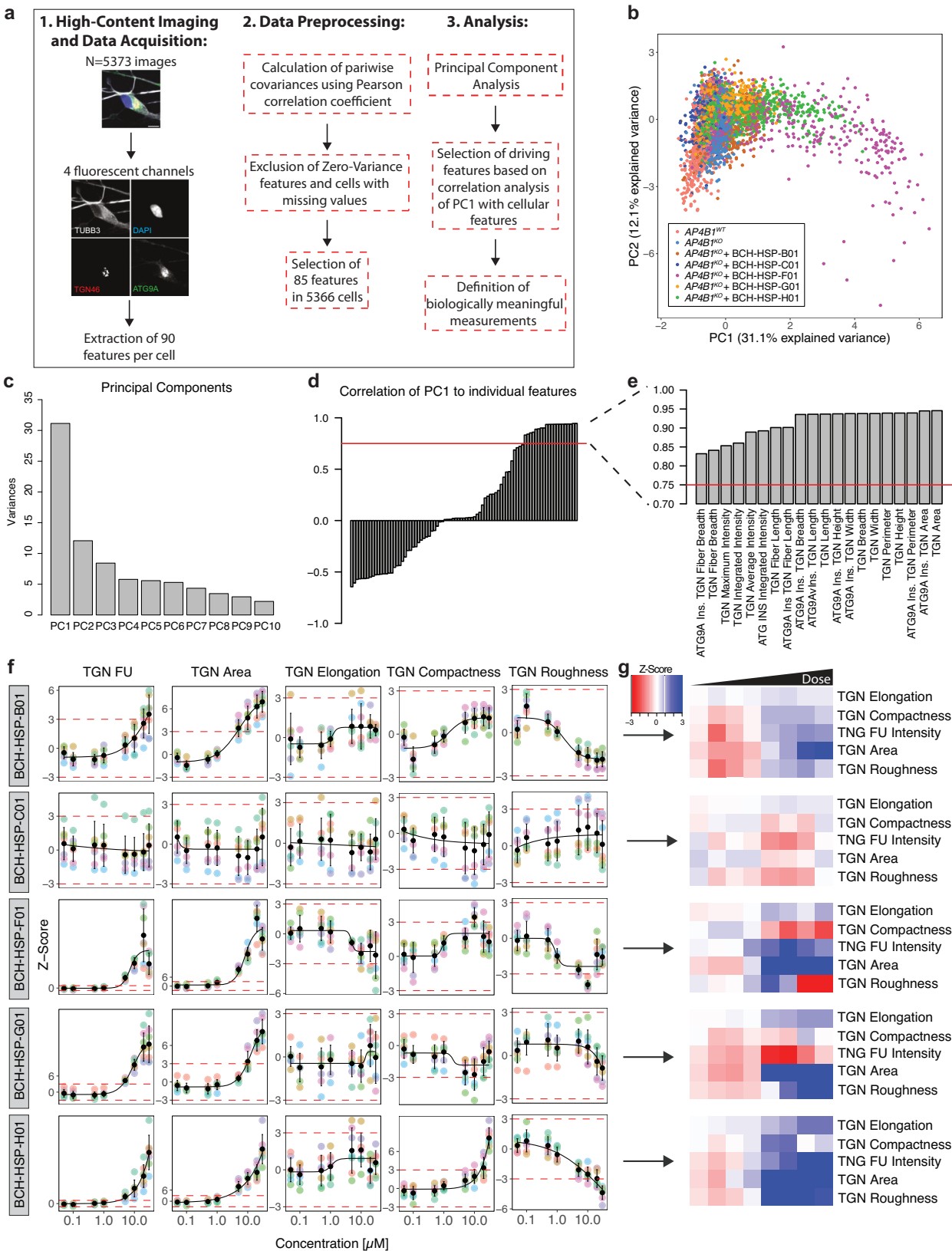

approximately 36% (Fig. 8l). Upon treatment with bafilomycin A1, both RAB3C knockout alone and combined knockout of RAB3C and RAB12 further increased BCH-HSP-C01-mediated LC3-I to LC3-II conversion (Fig. 8j–l). These findings suggest the possibility that RAB3C and RAB12 modulate BCH-HSP-C01-mediated ATG9A trafficking and subsequent autophagy induction.

## Discussion

Identification of therapeutic targets for rare neurological diseases represents a major scientific and public health challenge[1,4]. The increasing number of rare genetic diseases[54], the rising rate of diagnoses[55], and the significant burden for patients[56,57], caregivers[58] and healthcare systems[59] highlight the urgent need for translational

**Fig. 4 | Multiparametric profiling of 5 active compounds in *AP4B1^{KO}* SH-SY5Y cells. a** Multiparametric profiles of images of 5373 cells were acquired using 4 fluorescent channels. Scale bar: 10 μm. A total of 90 measurements per cell were generated for the cytoskeleton (beta-3 tubulin), the nucleus (DAPI), the TGN (TNG46) and ATG9A vesicles (ATG9A). The different steps of data preprocessing and phenotypic clustering using principal component analysis (PCA) are shown. **b** PCA shows different clusters of cells based on 85 phenotypic features. Experimental conditions are color-coded. The first two principal components (PC1 and PC2) explain 43.2% of the observed variance. **c** Bar plot summarizing the variance explained by the first 10 PCs. Most of the variance is explained by PC1 and, to a lesser degree, PC2. **d** Correlation analysis of PC1 with all 85 features using the Pearson correlation coefficient. The red dashed line represents a cutoff for correlations >0.75. **e** Zoom-in on selected features of interest showing a correlation with PC1 >0.75. **f** Measurements of TGN intensity and descriptors of TGN shape and network complexity for the individual hit compounds as line graphs. Data points represent per well means of 7 independent plates. Black dots and error bars represent mean ± 1 SD. **g** Information on TGN summarized using heatmap visualization.

research that moves beyond gene discovery to the identification of disease mechanisms and therapies. Unbiased high-content small molecule screens are a platform for drug-repurposing approaches and a starting point for the rationale development of new compounds[1–6]. Disease-relevant 'screenable' phenotypes across cellular models, including patient-derived cells, provide an entry point into developing automated, high-content screening and analysis platforms.

In this study, we develop the first high-throughput cell-based phenotypic screening platform for a prototypical form of childhood-onset HSP caused by defective protein trafficking. Our platform allows us to determine the subcellular localization of the AP-4 cargo protein ATG9A in several cellular models of AP-4-deficiency. The hypothesis that ATG9A mislocalization is a key mechanism in the pathogenesis of AP-4-HSP is supported by the independent work of the Robinson[12], Kittler[14] and Bonifacino[11,13,60] groups, in addition to our own work[15,21,24,25], and by the overlapping phenotypes of AP-4[13,14,26] and *Atg9a*[28] knockout mice.

ATG9A is the only conserved autophagy-related transmembrane protein[53], and in mammalian cells, cycles between the TGN and ATG9A vesicles, which associate with endosomes[61] and autophagosome formation sites[61,62]. ATG9A has 4 transmembrane domains and forms homotrimers that have lipid scramblase activity[51–53], postulated to equilibrate lipids in the double-membrane layer of nascent autophagosomes[63,64]. Basal levels of autophagy are essential for neuronal survival, and neuron-specific ablation of the autophagy pathway leads to axonal degeneration and cell death[65–67]. In neurons, autophagosomes form in the distal axon[68,69] and are subject to active transport[70–72]. Thus, efficient vesicular trafficking and spatial distribution of ATG9A are essential for axonal function as demonstrated in CNS-specific *Atg9a* knockout mice[28].

Having established a robust and dynamic assay that reliably measures intracellular ATG9A distribution, we systematically screened a large library of 28,864 small molecules for their ability to restore ATG9A trafficking from the TGN to the cytoplasm. Following this primary screen, a counter-screen and a series of orthogonal experiments identified a small molecule, termed BCH-HSP-C01, that can restore the intracellular distribution of ATG9A and a second transmembrane AP-4 cargo protein, DAGLB, in neuronal models of AP-4 deficiency, including hiPSC-derived neurons from two patients with AP-4-HSP.

Compound BCH-HSP-C01 has physicochemical properties that are within the parameters that are optimal for CNS drugs[73] and therefore represents a strong candidate for an in vivo tool compound. In addition, the low molecular weight and topological polar surface area create opportunities for compound optimization. Since the molecular targets of BCH-HSP-C01 are unknown, we employed a target deconvolution strategy using transcriptomics and proteomics to define the cellular pathways impacted by this small molecule. This approach identified two central themes: (1) modulation of Golgi dynamics and vesicular trafficking and (2) engagement of autophagy. At the core of the putative pathways affected by BCH-HSP-C01, we identified the RAB proteins RAB1B, RAB3C and RAB12, as well as the interacting RAB geranyl transferase subunits CHM and CHML. RAB3C and RAB12 showed the strongest and most consistent association with BCH-HSP-C01 treatment in both SH-SY5Y cells and hiPSC-derived

neurons, and our analyses suggest that these two proteins likely contribute to BCH-HSP-C01-mediated redistribution of ATG9A from the TGN and increase of autophagic flux.

RAB proteins comprise a large family of small guanosine triphosphate (GTP) binding proteins that act as key regulators of intracellular membrane trafficking in eukaryotic cells at several stages, including cargo sorting, vesicle budding, docking, fusion and membrane organization[74,75]. RAB GTPases function both as soluble and specifically localized integral-membrane proteins, the latter being mediated by prenylation. Among the roughly 70 known RAB proteins, more than 20 are primarily associated with the TGN, where they regulate Golgi organization, coordinate vesicle trafficking and interact with various steps of the autophagy pathway[49,50].

Following treatment with BCH-HSP-C01, the RAB protein family members RAB3C and RAB12 were consistently downregulated in both SH-SY5Y cells and hiPSC-derived neurons. Knockout experiments of these two proteins revealed that their loss potentiates BCH-HSP-C01-mediated ATG9A translocation and autophagic flux. RAB3C, which is part of the RAB3 superfamily, is primarily expressed in brain and endocrine tissues, where it localizes to the Golgi and synaptic vesicles and is involved in exocytosis and modulation of neurotransmitter release[76]. RAB12 is mainly localized to recycling endosomes, where it regulates endosomal trafficking and lysosomal degradation and has been identified as a modulator of autophagy[77]. A well-known downstream target of RAB12 is the transferrin receptor (TfR). Knockdown of RAB12 in mouse embryonic fibroblasts increases TfR protein levels, while overexpression leads to its reduction[48]. In line with this, we find that treatment with BCH-HSP-C01 reduced RAB12 protein levels while, at the same time, robustly elevating transferrin receptor protein 1 (TFRC). To the best of our knowledge, no interaction between RAB3C and RAB12 has been described so far; however, our data support the possibility that both proteins are involved in BCH-HSP-C01-mediated modulation of vesicle trafficking and autophagic flux.

Our study has identified the first candidate small molecule drug capable of restoring protein mislocalization in AP-4-deficient cells, including human neurons from patients. We acknowledge several limitations of our approach, some of which are inherent to high-throughput screens and some that are specific to our assay. First, in the primary screen, compounds were arrayed to single wells and only one well per compound was screened. We recognize that using multiple replicates as well as multiple concentrations and treatment durations could have potentially decreased the rate of false negative results. However, we prioritized efficiency and compounds that would show a robust impact at a single low-micromolar concentration. With respect to false positive results, these were eliminated in the counter-screen, which was performed with biological duplicates and using dose-response titrations covering a broad range of concentrations. Second, as ATG9A mislocalization is a cellular phenotype of AP-4 deficiency conserved in non-neuronal and neuronal cells both in vitro[11–15,25,78] and in vivo[13,14,27], we decided to conduct the initial screen in patient-derived fibroblasts, as a simple cellular model of AP-4 deficiency. While the use of patient fibroblasts in the primary screen increases translational relevance, compounds that would have the capacity to correct ATG9A trafficking exclusively in neuronal cells could be missed at this stage.

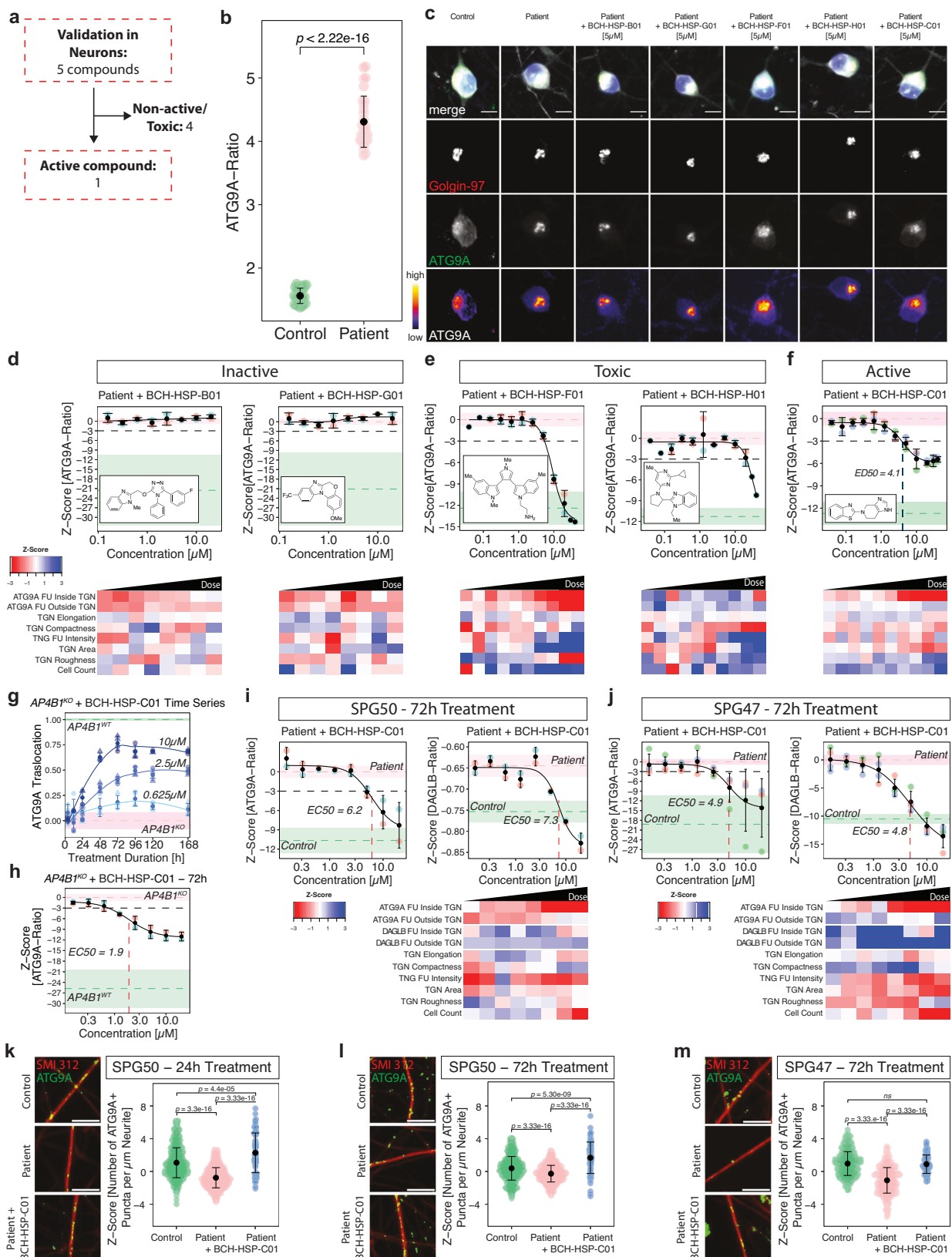

We determined that this risk was outweighed by the benefits of a robust assay performance and the fact that mechanisms of AP-4-mediated protein trafficking are conserved across tissues and cell types[11–15,35,78]. Third, even though cell-based disease models can, to some extent, mimic the complexity of therapeutic responses in biological systems, the translation to in vivo models is often challenging,

particularly for neurodevelopmental and neurodegenerative diseases. Considerations such as a lead compound's ability to cross the blood-brain-barrier, target engagement in the central nervous system, therapeutic responses in complex neuronal networks relying on interactions with glia cells, developmental windows amenable to therapy, as well as in vivo off-target effects and toxicity must be considered and

**Fig. 5 | BCH-HSP-C01 restores ATG9A and DAGLB trafficking in hiPSC-derived neurons from AP-4-HSP patients. a** Overview of 5 active compounds in hiPSC-derived cortical neurons from a patient with *AP4M1*-associated SPG50 compared to same-sex parent (heterozygous control). **b** Baseline differences of ATG9A ratios using per well means of 60 wells per condition from 5 plates. Means are shown as black dots; whiskers represent ±1.5 x IQR. Statistical testing was done using the Mann-Whitney U test. *P*-values are two-sided. **c** Representative images of hiPSC neurons treated with individual compounds at 5 μM for 24 h. Scale bar: 10 μm. **d**–**f** Dose-response curves for ATG9A ratios in hiPSC-derived neurons from a patient with SPG50 treated with individual compounds for 24 h, along with their morphological profiles depicted as heatmaps. All data points represent per well means of 2 (d, e), or 4 (f) independent differentiations. Black dots and error bars represent mean ± 1 SD. Dashed lines show mean *Z*-scores for positive (green) and negative (salmon) controls. Shaded areas represent ± 1 SD. **g** Time-series experiment of *AP4B1*[KO] SH-SY5Y cells treated with BCH-HSP-C01 with different concentrations and treatment durations. Data points represent per well means of two independent plates. Shapes indicate technical replicates. Dashed lines show mean *Z*-scores for positive (green) and negative (salmon) controls. Shaded areas represent ± 1 SD. **h** Dose-response curve for ATG9A ratios in *AB4B1*[KO] SH-SY5Y cells

treated with BCH-HSP-C01 for 72 h. Data points represent per well means from two independent plates. Dashed lines show mean *Z*-scores for positive (green) and negative (salmon) controls. Shaded areas represent ± 1 SD. **i, j** Dose-response curves for ATG9A and DAGLB ratios in hiPSC-derived neurons from a patient with SPG50 (**i**) and SPG47 (**j**) after prolonged treatment with BCH-HSP-C01 for 72 h, along with morphologic profiles. Data points represent per well means of 2 independent differentiations. Dashed lines show mean *Z*-scores for positive (green) and negative (salmon) controls. Shaded areas represent ± 1 SD. **k**–**m** Quantification of ATG9A positive puncta per neurite length in hiPSC-derived neurons from a patient with SPG50 following 24 h of BCH-HSP-C01 treatment (**k**, $n_{WT/LoF}$ = 837, $n_{LoF/LoF}$ = 848, $n_{LoF/LoF + BCH-HSP-C01}$ = 72), as well as hiPSC-derived neurons from patients with SPG50 (**l**, $n_{WT/LoF}$ = 843, $n_{LoF/LoF}$ = 843, $n_{LoF/LoF + BCH-HSP-C01}$ = 70) and SPG47 (**m**, $n_{WT/LoF}$ = 424, $n_{LoF/LoF}$ = 428, $n_{LoF/LoF + BCH-HSP-C01}$ = 70) treated for 72 h with BCH-HSP-C01. All data points represent means of single images from two independent differentiations. Statistical testing was done using the Mann-Whitney U test. *P*-values are two-sided and were adjusted for multiple testing using the Benjamini-Hochberg procedure. Representative images are shown for all experimental conditions. Scale bar: 10 μm.

explored in future studies. To mitigate some of these risks, we employed unbiased multiparametric profiling of BCH-HSP-C01, which suggested little off-target effects. Future studies are required to exclude pleiotropic effects or off-target toxicity in different cell types or tissues in vivo. Lastly, while BCH-HSP-C01 leads to a redistribution of two well-established AP-4 cargo proteins, ATG9A and DAGLB, we are unable to exclude the possibility that other neuron-specific cargos of AP-4 exist and are important for the pathogenesis of AP-4-HSP. Nonetheless, mislocalization of both proteins is proposed as the major contributor to neuronal pathology caused by AP-4 deficiency through dysregulated autophagy and endocannabinoid signaling, respectively[11–14,29]. Our automated high-throughput platform would allow for the rapid interrogation of additional AP-4 cargo proteins in the future.

In conclusion, our findings provide a solid foundation for lead optimization of BCH-HSP-C01 and its development as a potential therapeutic, with the next step being in vivo proof-of-concept experiments. More broadly, our approach illustrates the development of a small molecule screening platform for a rare neurogenetic disease, leveraging robust cellular phenotypes. We hope this approach will create a paradigm for other rare and more common disorders of protein trafficking. The increase of autophagic flux through BCH-HSP-C01 offers the intriguing possibility that this compound could be considered for the treatment of other autophagy-associated diseases.

## Methods
### Clinical data from patients with AP-4-HSP
This study was approved by the Institutional Review Board at Boston Children's Hospital (IRB-P00033016 and IRB-P00016119). Two patients with AP-4-HSP and their clinically-unaffected, sex-matched parents were enrolled in the International Registry and Natural History Study for Early-Onset Hereditary Spastic Paraplegia (ClinicalTrials.gov Identifier: NCT04712812). Both patients had a clinical and molecular diagnosis of AP-4-HSP and presented with core clinical and imaging features[8]. Patient 1 (2-year-old male) was diagnosed with *AP4B1*-associated SPG47 and carries the following compound-heterozygous variants: NM_001253852.3: c.1160_1161del (p.Thr387ArgfsTer30) / c.1345A>T (p.Arg449Ter). The sex-matched parent (38 years old) carries the heterozygous c.1160_1161del; p.Thr387Argfs*30 variant. Patient 2 (18-month-old male) was diagnosed with *AP4M1*-associated SPG50 and carries the following compound-heterozygous variants: NM_004722.4: c.916C>T (p.Arg306Ter) / c.694dupG (p.Glu232GlyfsTer21). The sex-matched parent (40 years old) carries the heterozygous c.694dupG (p.Glu232GlyfsTer21) variant.

### Antibodies and reagents
The following reagents were used: Bovine serum albumin (American-BIO, Cat# 9048-46-8), saponin (Sigma, #47036-50G-F), normal goat serum (Sigma-Aldrich, Cat# G9023-10ML), Dulbecco's phosphate-buffered saline (DPBS) (Thermo Fisher Scientific, Cat# 14190-250), trypsin (Thermo Fisher Scientific, Cat#25200056), 4% paraformaldehyde (4%) (Boston BioProducts, Cat# BM-155), dimethyl-sulfoxide (DMSO) (American Bioanalytical, Cat# AB03091-00100), bafilomycin A1 (Enzo Life Sciences, Cat#BML-CM110-0100), chloroquine (Med-ChemExpress, Cat# HY-17589A), Molecular Probes Hoechst 33258 (Thermo Fisher Scientific, Cat# H3569) and Alexa Fluor 647-labeled phalloidin (Thermo Fisher Scientific, Cat#A22287). The following primary antibodies were used: Anti-AP4E1 at 1:500 (BD Bioscience, Cat# 612019), anti-ATG9A at 1:500-1000 (Abcam, Cat# ab108338), anti-DAGLB at 1:500 (Abcam, Cat# 191159), anti-TGN46 at 1:800 (Bio-Rad, Cat# AHP500G), anti-Golgin-97 1:500 (Abcam, Cat# 169287), anti-beta-Tubulin III 1:1000 (Synaptic Systems, Cat# 302304 and Sigma, Cat# T8660), anti-beta-Actin 1:10,000 (Sigma, Cat# A1978-100UL), anti-SMI 312 (Biolegend, Cat # 837904), anti-pan-AKT (Cell Signaling Technology, Cat# 4691), anti-Histon H3 (Cell Signaling Technology, Cat # 9715), anti-RAB12 (Santa Cruz, Cat# sc-515613), anti-RAB3C (Santa Cruz, Cat# 107 203), anti-LC3B 1:1000 (Novus, Cat#100-2220). Fluorescently labeled secondary antibodies for immunocytochemistry were used at 1:2000 (Thermo Fisher Scientific, Cat# A11005, A-11008, A-11016, A-11073, A-21235, A-21245), for western blotting at 1:5000 (LI-COR Biosciences, Cat# 926-68022, 926-68023, 926-32212, 926-32213).

### Small molecule library
A diversity small molecule library containing 28,864 compounds was provided by Astellas Pharma Inc. Compounds were arrayed in 384-well microplates at a final concentration of 10 mM (1000-fold the screening concentration) in DMSO. Assay plates were stored at −80 °C and thawed 30 min prior to cell plating. Active compounds from the primary screen were re-screened in a secondary screen, using eleven-point concentrations (range: 0.04 μM, 0.08 μM, 0.16 μM, 0.31 μM, 0.63 μM, 1.25 μM, 2.5 μM, 5 μM, 10 μM, 20 μM, 40 μM) in two biological replicates. The chemical structures of the 5 compounds that were tested in neuronal models were disclosed by Astellas Pharma Inc. after the screen was completed.

### Fibroblast cell culture
Fibroblast lines were collected from individuals enrolled in our natural history study (approved at Boston Children's Hospital, IRB-P00033016). Probands provided written consent for routine skin punch biopsies. Fibroblasts were derived from both patients and their

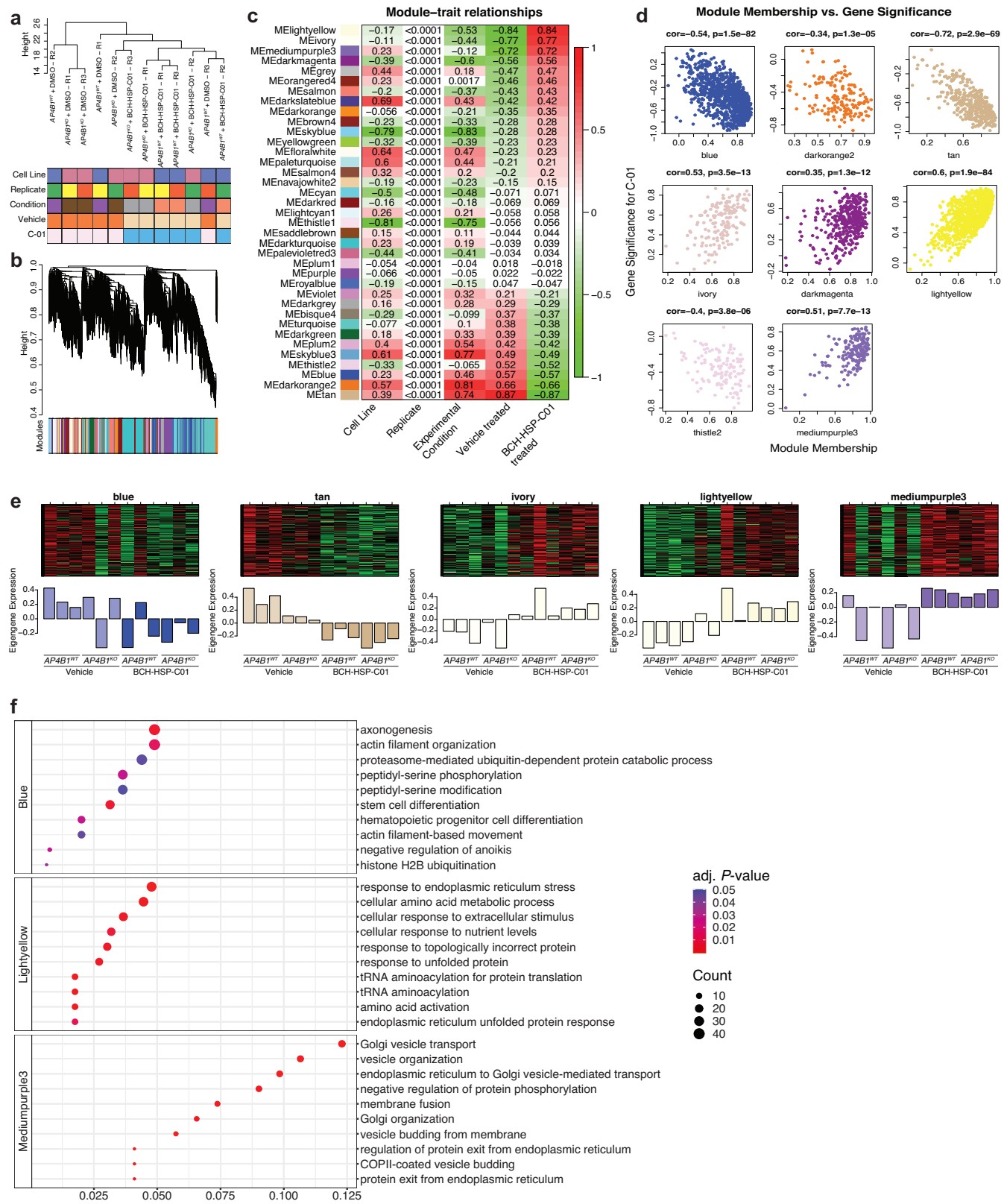

respective sex-matched heterozygous parents[15]. Primary human skin fibroblasts were cultured and maintained as previously described[79]. Briefly, cells were maintained in DMEM high glucose (Gibco, #11960044) supplemented with 20% FBS (Gibco, #10082147), penicillin 100 U/mL and streptomycin 100 μg/mL (Gibco, #15140122). Cells were kept in culture for up to 8 passages and routinely tested for the presence of mycoplasma contamination. For high-throughput imaging, fibroblasts were seeded onto 384-well plates (Greiner Bio-One, #781090) at a density of $2 \times 10^3$ per well using the Multidrop Combi Reagent Dispenser (Thermo Fisher Scientific, #11388-558). Media changes were done every 2–3 days and drugs were administered 24 h before fixation.

**Fig. 6 | Target deconvolution using bulk RNA sequencing and weighted gene co-expression network analysis in *AP4B1^(KO)* SH-SY5Y cells treated with BCH-HSP-C01. a** Hierarchical clustering of 12 samples using average linkage showed two main clusters based on treatment with vehicle vs. BCH-HSP-C01, irrespective of cell line. **b** Cluster dendrogram of 18,506 expressed genes based on topological overlap. Clusters of co-expressed genes ("modules") were isolated using hierarchical clustering and adaptive branch pruning. **c** Heatmap visualization of the correlation of gene expression profiles ("module eigengene", ME) of each module with measured traits. Pearson correlation coefficients are shown for each cell of the heatmap. **d** Intramodular analysis of module membership (MM) and gene significance (GS) for highly correlated modules, allowing identification of genes that have high significance with treatment as well as high connectivity to their modules. Statistical testing was done using the *t*-test. *P*-values are two-sided. **e** ME expression profiles for the top 5 co-expressed modules. **f** Gene ontology enrichment analysis showed enriched pathways in 3/5 modules. Statistical testing was done using the hypergeometric test. *P*-values are one-sided. Pathways were considered differentially expressed with an FDR <0.05.

## SH-SY5Y cell culture

*AP4B1* wildtype (*AP4B1^(WT)*) and *AP4B1* knockout (*AP4B1^(KO)*) SH-SY5Y cells were generated previously[12]. Undifferentiated SH-SY5Y cells were maintained in DMEM/F12 (Gibco, Cat# 11320033) supplemented with 10% heat-inactivated fetal bovine serum (Gibco, Cat# 10438026), 100 U/mL penicillin and 100 μg/mL streptomycin at 37 °C under 5% CO_2. SH-SY5Y cells were passaged every 2–3 days and differentiated into a neuron-like state using a 5-day differentiation protocol with all-*trans*-retinoic acid (MedChemExpress, #HY-14649) as described previously[33]. For assessment of ATG9A translocation, differentiated SH-SY5Y cells were plated in 96-well plates (Greiner Bio-One, Cat# 655090) at a density of $1 \times 10^4$ cells per well. Media changes were done every 2–3 days and drugs were administered 24–72 h before fixation.

## Generation of hiPSC lines and neuronal differentiation

Fibroblasts were reprogrammed to hiPSCs using non-integrating Sendai virus as described previously[35,36]. Quality control experiments, including karyotyping, embryoid body formation, pluripotency marker expression, STR profiling and Sanger sequencing for *AP4B1* or *AP4M1* variants, were reported previously[35,36]. hiPSC-derived neurons were generated using induced NGN2 expression following published protocols with minor modifications[37,38]. hiPSCs were dissociated into single cells with accutase (Innovative Cell Technology, Cat#AT 104–500) and seeded onto Geltrex-coated plates (Thermo Fisher Scientific, Cat#A1413301). hiPSCs were then infected with concentrated rtTA-and NGN2-expressing lentiviruses (FUW-M2rtTA Addgene #20342, pTet-O-Ngn2-puro Addgene #52047) in the presence of polybrene (8 μg/mL, Sigma-Aldrich, Cat# TR-1003-G). The next day, hiPSCs were fed with supplemented mTeSRPlus and expanded for cryopreservation. In parallel, a kill curve was generated to determine the optimal puromycin concentration needed to eliminate untransduced cells. Successful transduction was established by adding doxycycline (2 μg/mL, Millipore, Cat#324385–1GM) to virus-treated cells for 24 h, followed by adding the optimized puromycin concentration (Invitrogen, 1 μg/mL, Cat# ant-pr-1) for up to 48 h.

For the generation of glutamatergic neurons, NGN2 transduced hiPSCs were dissociated into single cells using accutase and seeded onto geltrex-coated plates. The following day, NGN2 expression was induced using doxycycline and selected with puromycin. Growth factors BDNF (10 ng/mL, Peprotech, Cat#450–02), NT3 (10 ng/mL, Peprotech, Cat# 450–03), and laminin (0.2 mg/L, Thermo Fisher Scientific, Cat#23017–015) were added in N2 medium for the first 2 days. Cells were then fed with BDNF (10 ng/mL), NT3 (10 ng/mL), laminin (0.2 mg/Lf), doxycycline (2 μg/mL), and Ara-C (4uM, Sigma-Aldrich, Cat# C1768) in B27 media every other day until differentiation day 6. On day 6, cells were dissociated with papain (Worthington, Cat# LK003178) and DNaseI (Worthington, Cat# LK003172) and replated on poly-D-lysine (0.5 mg/mL; Sigma-Aldrich, Cat#P6407) and laminin (5 μg/mL; Thermo Fisher Scientific, Cat #23017-015) coated plates either in co-culture with hiPSC-derived astrocytes (Astro.4U, Ncardia) for immunocytochemistry experiments, or without astrocytes for preparation of cell lysates for proteomics experiments. For assessment of ATG9A translocation, neurons were plated in 96-well plates at a density of $4 \times 10^4$ cells per well. Media changes were done every 2–3 days and drugs were administered 24–72 h before fixation.

## Immunocytochemistry

The immunocytochemistry workflow was optimized for high-throughput using automated pipettes and reagent dispensers (Thermo Fisher Scientific Multidrop Combi Reagent Dispenser, Integra VIAFLO 96/384 liquid handler, Integra VOYAGER pipette). Fibroblasts and SH-SY5Y cells were fixed using 3% and 4% PFA, respectively, permeabilized with 0.1% saponin in PBS and blocked in 1% BSA/0.01% saponin (blocking solution) in PBS. hiPSC-derived neurons were fixed in 4% PFA and permeabilized and blocked using 0.1% Triton X-100/2% BSA/0.05% NGS in PBS. Primary antibody (diluted in blocking solution) was added for 1 h (fibroblasts and SH-SY5Y cells) at room temperature or overnight (hiPSC neurons) at 4 °C. Plates were gently washed three times in blocking solution (fibroblasts and SH-SY5Y cells) or in PBS (hiPSC neurons), followed by addition of fluorochrome-conjugated secondary antibodies, Hoechst 33258 and phalloidin for 30 min (fibroblasts) or Hoechst 33258 for 60 min (SH-SY5Y cells and hiPSC neurons) at room temperature. Plates were then gently washed three times with PBS and protected from light.

## High-content imaging and automated image analysis

High-throughput confocal imaging was performed on the ImageXpress Micro Confocal Screening System (Molecular Devices) using an experimental pipeline modified from the pipeline described in Behne et al.[15]. For experiments in fibroblasts, images were acquired using a 20x S Plan Fluor objective (NA 0.45 μM, WD 8.2–6.9 mm). Per well, 4 fields were acquired in a 2 × 2 format (384-well plates). For experiments in SH-SY5Y cells and hiPSC neurons, up to 36 fields were acquired in a 6 × 6 format (96-well plate) using a 40x S Plan Fluor objective (NA 0.60 μm, WB 3.6–2.8 mm). Image analysis was performed using a customized image analysis pipeline in MetaXpress (Molecular Devices): Briefly, cells were identified based on the presence of DAPI signal inside a phalloidin (fibroblasts) or TUBB3 (SH-SY5Y cells and hiPSC neurons)-positive cell body. Sequential masks were generated for (1) the TGN by outlining the area covered by TGN marker TGN46 (TGN46-positive area, in fibroblasts and SH-SY5Y cells) or Golgin 97 (Golgin 97-positive area, in hiPSC neurons) and (2) for the cell area outside the TGN (actin-positive area minus TGN46-positive area). ATG9A fluorescence intensity (F.U.) was measured in both compartments in each cell, and the ATG9A ratio was calculated by dividing the ATG9A fluorescence intensity inside the TGN by the ATG9A fluorescence intensity in the remaining cell body (Fig. 1b):

$$ATG9A\ Ratio = \frac{ATG9A\ F.U.\ inside\ the\ TGN}{ATG9A\ F.U.\ outside\ the\ TGN} \quad (1)$$

Additional masks for the TGN used for morphologic profiling included TGN Roughness (shape factor in the MetaXpress software) and the following calculated metrics:

$$TGN\ Elongation = \frac{TGN\ With}{TGN\ Length} \quad (2)$$

$$TGN\ Compactness = \frac{(TGN\ Perimeter)^2}{4\pi * TGN\ Area} \quad (3)$$

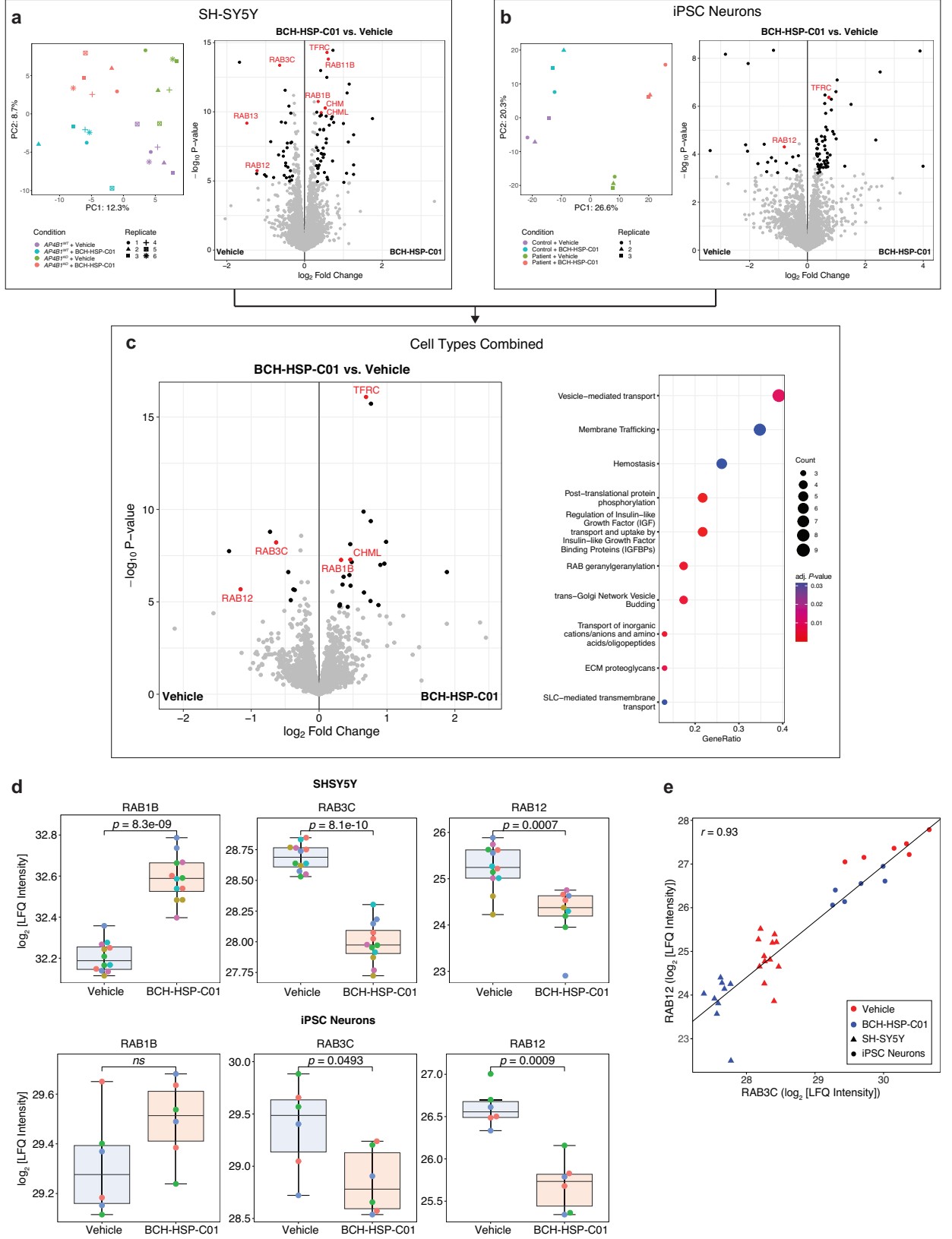

Z′-factor robust values and strictly standardized median difference (SSMD)[30] were calculated for each plate and only plates that met the predefined quality metrics of a Z′-factor robust ≥0.3 and SSMD ≥3 were included in subsequent analyses.

## Western blotting

Western blotting was done as previously described[72]. Briefly, cells were lysed in RIPA lysis buffer (Thermo Fisher Scientific Cat# 89900) supplemented with cOmplete protease inhibitor (Roche Cat#

**Fig. 7 | Target deconvolution using unbiased quantitative proteomics in *AP4B1^KO* SH-SY5Y cells and AP-4-HSP patient-derived hiPSC neurons treated with BCH-HSP-C01. a–c** Differential protein enrichment analysis. Statistical testing was done using protein-wise linear models and empirical Bayes statistics using the limma package in R (Ritchie et al.). Proteins were considered as differentially enriched with an FDR <0.05 and a log$_2$ fold change >0.3. PCA plots show the top 500 variable proteins. Differentially enriched proteins are shown in volcano plots colored in black. Proteins with the most consistent enrichment profiles across all experimental conditions (see Supplementary Fig. 8) are colored and labeled in red. **a** SH-SY5Y cells: 8141 unique proteins were analyzed. **b** hiPSC-derived neurons: 7386 unique proteins were analyzed. **c** Integrated analysis of SH-SY5Y cells and hiPSC-derived neurons: 5357 unique proteins were analyzed. The dot plot summarizes dysregulated Reactome pathways of the pooled analysis. Pathways were considered differentially expressed with an FDR <0.05. **d** The RAB protein family

members RAB1B, RAB3C and RAB12 showed the most consistent profiles in response to BCH-HSP-C01 treatment and were selected for further analysis. LFQ intensities in SH-SY5Y cells (*AP4B1^WT* and *AP4B1^KO* pooled; 12 independent experiments per condition; exception: RAB12 in BCB-HSP-C01 treated SH-SY5Y cells was not detectable in 3 samples, which is why quantification is based on 9 independent experiments), and hiPSC-derived neurons (controls and patients pooled, 6 independent experiments per condition) are shown. Box plots show medians (center), upper and lower quartiles (hinges) and 1.5 x IQR (whiskers). Statistical testing was done using pairwise *t*-tests. *P*-values are two-sided and were adjusted for multiple testing using the Benjamini-Hochberg procedure. **e** Correlation of LFQ intensities of RAB3C and RAB12 in *AP4B1^WT* (*n* = 11 samples) and *AP4B1^KO* (*n* = 10 samples) SH-SY5Y cells, as well as control (*n* = 6 samples) and patient (*n* = 6 samples) hiPSC-derived neurons are measured by the Pearson correlation coefficient (*r*).

---

04693124001) and PhosSTOP phosphatase inhibitor (Roche Cat# 4906845001). Total protein concentration was determined using a Pierce BCA Protein Assay Kit (Thermo Fisher Scientific, Cat# 23225). Equal amounts of protein were solubilized in LDS sample buffer (Thermo Fisher Scientific, Cat# NP0008) under reducing conditions, separated by gel electrophoresis, using 4–12% (Thermo Fisher Scientific, Cat# NW04125BOX) or 12% Bis-Tris gels (Thermo Fisher Scientific, Cat# NP0343BOX) and MOPS or MES buffer (Thermo Fisher Scientific, #NP0001 and #NP0002) and transferred to PVDF or nitrocellulose membranes (EMD Millipore, #SLHVR33RS). Following blocking with blocking buffer (LI-COR Biosciences, #927-70001), membranes were incubated overnight with the respective primary antibodies. Near-infrared fluorescent-labeled secondary antibodies (IR800CW, IR680LT; LI-COR Biosciences) were used and quantification was done using the Odyssey infrared imaging system and Empiria Studio Software (LI-COR Biosciences).

## Multiparametric morphological profiling

The multiparametric morphological profiling strategy employed in this study was adapted from previously published protocols[34]. Single-cell measurements of ninety distinct descriptors of shape and intensity for the nucleus (DAPI), the cytoskeleton and global cell morphology (anti-beta-Tubulin III), the TGN (anti-TGN46), and ATG9A vesicles (anti-ATG9A) were acquired and automatically extracted. Single-cell data were summarized by computing per-image medians for each variable (Source Data file 6). Next, a correlation matrix was generated using the Pearson correlation coefficient with complete pairwise observations. Variables with zero variance and observations with missing values were removed. Variables were then transformed to have a mean of zero and a standard deviation of one. Principal component analysis (PCA) was conducted to reduce dimensionality and cluster data based on their properties. To identify the contribution of individual features to the variance within the dataset, correlation analysis was performed between the first principal component, accounting for the majority of the variance within the dataset, and all extracted features. Features displaying a correlation coefficient >0.75 were selected to define morphological profiles. Profiles were summarized using heatmap visualization.

## Sample preparation for RNA extraction

SH-SY5Y cells were differentiated with retinoic acid as described above and subsequently treated with compounds of interest for 72 h prior to lysis using Qiagen RTL-Buffer supplemented with 1% ß-mercaptoethanol. RNA extraction, library preparation and sequencing were conducted at Azenta Life Sciences (South Plainfield, NJ, USA). Total RNA was extracted from frozen cell pellet samples using a Qiagen Rneasy mini kit following the manufacturer's instructions (Qiagen, Cat# 74004).

## Library preparation with polyA selection and Illumina sequencing

RNA samples were quantified using a Qubit 4 Fluorometer (Life Technologies), and RNA integrity was checked using Agilent TapeStation 4200 (Agilent Technologies). RNA sequencing libraries were prepared using the NEBNext Ultra II RNA Library Prep Kit for Illumina using the manufacturer's instructions (New England Biolabs). Briefly, mRNAs were initially enriched with Oligod(T) beads. Enriched mRNAs were fragmented for 15 min at 94 °C. First-strand and second-strand cDNA were subsequently synthesized. cDNA fragments were end-repaired and adenylated at 3' ends, and universal adapters were ligated to cDNA fragments, followed by index addition and library enrichment by PCR with limited cycles. The sequencing library was validated on the Agilent TapeStation (Agilent Technologies) and quantified using Qubit 4 Fluorometer (Invitrogen) as well as by quantitative PCR (KAPA Biosystems). The sequencing libraries were clustered on 3 lanes of a flowcell. After clustering, the flowcell was loaded on the Illumina instrument (HiSeq 4000 or equivalent) according to the manufacturer's instructions. The samples were sequenced using a 2x150bp Paired End (PE) configuration. Image analysis and base calling were conducted by the Control software. Raw sequence data (.bcl files) generated by the sequencer were converted into fastq files and demultiplexed using Illumina's bcl2fastq 2.17 software. One mismatch was allowed for index sequence identification.

## Downstream RNA sequencing analysis

Sequencing reads were mapped to the GRCh38 reference genome available on ENSEMBL using the STAR aligner v.2.7.9a. Differential expression analysis was done using the TREAT approach developed by McCarthy and Smyth[80], implemented in the edgeR package in R. Raw counts were obtained using STAR, and low expressed genes were excluded using the method described by Chen et al.[81]. Expression data were normalized using the Trimmed Mean of M-values method implemented in the edgeR package. Genes were considered as differentially expressed according to default options with a false discovery rate (Benjamini-Hochberg procedure) <0.05 and a log$_2$ fold change of >0.3. Gene ontology (GO) enrichment analysis was done using clusterProfiler[82]. Pathways were considered differentially expressed with an FDR <0.05.

## Network connectivity analysis

To identify transcriptional changes in co-expressed groups of genes following compound treatment, a weighted gene co-expression network analysis (WGCNA) was performed. Raw counts were generated, and low-expressed genes were removed as described above. Data were normalized using variance stabilizing transformation as described by Anders et al.[83]. Batch effects were removed using the limma package in R[84]. Preprocessed data were then analyzed using the WGCNA package in R[85,86]. In brief, pairwise Pearson correlations were calculated between all genes, and genes with a positive correlation were selected

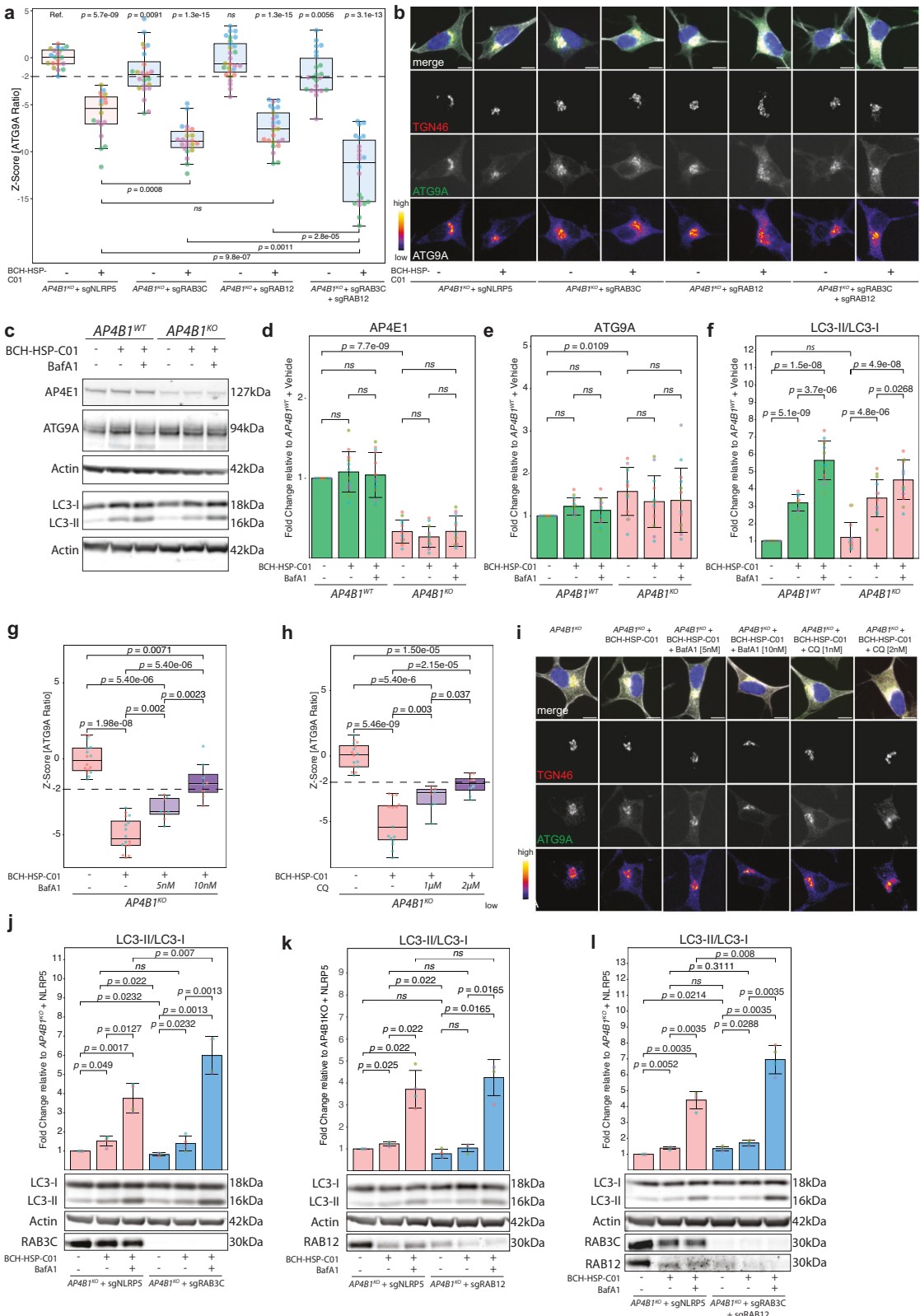

to form a "directed" correlation matrix. Next, the correlations were raised to a power to approximate a scale-free network. The adequate power was chosen based on soft thresholding aiming for a high scale independence above 0.8 by keeping a mean connectivity between 200 and 500. Genes were then grouped based on topological overlap and clusters were isolated using hierarchical clustering and adaptive branch pruning of the hierarchical cluster dendrogram, giving rise to

groups of co-expressed genes, so-called modules. Gene expression profiles within each module were summarized using the "module eigengene" (ME), defined as the first principal component of a module. Within each module, the association of MEs with measured clinical traits was examined by correlation analysis. For these selected modules, module eigengene-based connectivity was determined for every gene by calculating the absolute value of the Pearson correlation

**Fig. 8 | RAB3C and RAB12 are involved in BCH-HSP-C01-mediated vesicle trafficking and enhancement of autophagic flux. a** *AP4B1*$^{KO}$ SH-SY5Y cells transfected for 72 h with RNPs targeting *RAB3C*, *RAB12* or both compared to *NLRP5* (non-essential control). Vehicle vs. BCH-HSP-C01 treatment at 5 μM was administered for 24 h. Data points represent per well means. Each experimental condition was tested in multiple replicates (n$_{AP4B1KO + sgNLRP5}$: 20 wells from 5 independent plates; n$_{AP4B1KO + sgNLRP5 + BCH-HSP-C01}$: 18 wells from 5 independent plates; n$_{AP4B1KO + sgRAB3C}$: 24 wells from 5 independent plates; n$_{AP4B1KO + sgRAB3C + BCH-HSP-C01}$: 22 wells from 5 independent plates; n$_{AP4B1KO + sgRAB12}$: 28 wells from 5 independent plates; n$_{AP4B1KO + sgRAB12 + BCH-HSP-C01}$: 25 wells from 5 independent plates; n$_{AP4B1KO + sgRAB3C + sgRAB12}$: 22 wells from 3 independent plates; n$_{AP4B1KO + sgRAB3C + sgRAB12 + BCH-HSP-C01}$: 22 wells from 3 independent plates). Statistical testing was done using the *t*-test. *P*-values are two-sided. **b** Representative images. Scale bar: 10 μm. **c** Representative western blots. Cells were treated with vehicle vs. BCH-HSP-C01 at 5 μM for 72 h. **d**–**f** Quantification of western blots. Experiments were performed in four biological replicates. Error bars represent ± 1 SD. Statistical testing was done using the *t*-test. *P*-values are two-sided and were adjusted for multiple testing using the Benjamini-Hochberg procedure. **g**, **h** *AP4B1*$^{KO}$ SH-SY5Y cells treated with BCH-HSP-C01 (5 μM) were incubated with ascending non-toxic doses of bafilomycin A1 (5 nM or 10 nM) or chloroquine (1 μM or 2 μM) for 24 h. Each condition was tested in 16 wells from 2 independent plates. (*AP4B1*$^{KO}$). **i** Representative images. Scale bar: 10 μm. **j**–**l** Representative western blots and quantification of whole cell lysates of *AP4B1*$^{KO}$ SH-SY5Y cells transfected for 72 h with RNPs against *RAB3C*, *RAB12* or both, compared to *NLRP5*. Vehicle vs. BCH-HSP-C01 treatment was administered for 48 h. Error bars represent ± 1 SD. Statistical testing was done using the *t*-test. *P*-values are two-sided and were adjusted for multiple testing using the Benjamini-Hochberg procedure. Box plots in all experiments show medians (center), upper and lower quartiles (hinges) and 1.5 x IQR (whiskers). Dashed lines represent a reduction of the ATG9A ratio of −2 SD compared to negative controls.

between the expression of the gene and the respective ME, producing a quantitative measure of module membership (MM). Similarly, the correlation of individual genes with the trait of interest was computed, defining gene significance (GS). Using the GS and MM, an intramodular analysis was performed, allowing identification of genes that have high significance with treatment as well as high connectivity to their modules. The biological information contained in modules of interest was summarized with gene ontology (GO) enrichment analysis using clusterProfiler[82]. Pathways were considered differentially expressed with a FDR <0.05.

## Sample preparation for mass spectrometry

Cells were lysed for whole proteome analysis in RIPA lysis buffer (Thermo Fisher Scientific, Cat# 89900) supplemented with cOmplete protease inhibitor (Roche Cat# 04693124001) and PhosSTOP phosphatase inhibitor (Roche Cat# 4906845001) and sonicated in a Bioruptor® Pico Sonication System (one single 30 s on/off cycle at 4 °C). Protein concentrations were determined using a Pierce BCA Protein Assay Kit (Thermo Fisher Scientific Cat# 23225). Lysates were stored at −80 °C until further processing. To generate peptide samples for analysis by mass spectrometry, 30–50 μg protein was precipitated by overnight incubation in 5 volumes of ice-cold acetone at −20 °C and pelleted by centrifugation at 10,000×*g* for 5 min at 4 °C. All subsequent steps were performed at room temperature. Precipitated protein pellets were air-dried, resuspended for denaturation and reduction in digestion buffer (50 mM Tris pH 8.3, 8 M Urea, 1 mM dithiothreitol (DTT)) and incubated for 15 min. Proteins were alkylated by the addition of 5 mM iodoacetamide for 20 min in the dark. Following reduction and alkylation, proteins were enzymatically digested by the addition of LysC (1 μg per 50 μg of protein; Wako, Cat# 129-02541) for overnight incubation. Samples were then diluted four-fold with 50 mM Tris pH 8.3 before the addition of Trypsin (1 μg per 50 μg of protein; Sigma-Aldrich, Cat# T6567) for 3 h. The digestion reaction was stopped by the addition of 1% (v/v) trifluoroacetic acid (TFA) and samples were incubated on ice for 5 min to precipitate contaminants, which were pelleted by centrifugation at 10,000×*g* for 5 min. Acidified peptides were transferred to new tubes before purification by solid-phase extraction using poly(styrenedivinylbenzene) reverse-phase sulfonate (SDB-RPS; Sigma-Aldrich, Cat# 66886-U) StageTips[87]. StageTips with three SDB-RPS plugs were washed with 100% acetonitrile, equilibrated with StageTip equilibration buffer (30% [v/v] methanol, 1% [v/v] TFA), and washed with 0.2% (v/v) TFA. 20 μg of peptides in 1% TFA were then loaded onto the activated StageTips, washed with 100% isopropanol, and then 0.2% (v/v) TFA. Peptides were eluted in three consecutive fractions by applying a step gradient of increasing acetonitrile concentrations: 20 μL SDB-RPS-1 (100 mM ammonium formate, 40% [v/v] acetonitrile, 0.5% [v/v] formic acid), then 20 μL SDB-RPS-2 (150 mM ammonium formate, 60% [v/v] acetonitrile, 0.5% [v/v] formic acid), then 30 μL SDB-RPS-3 (5% [v/v] NH4OH, 80% [v/v] acetonitrile). Eluted

peptides were dried in a centrifugal vacuum concentrator, resuspended in Buffer A* (0.1% (v/v) TFA, 2% (v/v) acetonitrile), and stored at −20 °C until analysis by mass spectrometry.

## Mass spectrometry

Mass spectrometry was performed on an Exploris 480 mass spectrometer coupled online to an EASY-nLC 1200 via a nano-electrospray ion source (all from Thermo Fisher Scientific). Per sample, 250 ng of peptides were loaded on a 50 cm by 75 μm inner diameter column, packed in-house with ReproSil-Pur C18-AQ 1.9 μm silica beads (Dr Maisch GmbH). The column was operated at 50 °C using an in-house manufactured oven. Peptides were separated at a constant flow rate of 300 nL/min using a linear 110 min gradient employing a binary buffer system consisting of Buffer A (0.1% [v/v] formic acid) and Buffer B (80% acetonitrile, 0.1% [v/v] formic acid). The gradient ran from 5 to 30% B in 84 min, followed by an increase to 60% B in 8 min, a further increase to 95% B in 4 min, a constant phase at 95% B for 4 min, and then a washout decreasing to 5% B in 5 min, before re-equilibration at 5% B for 5 min. The Exploris 480 was controlled by Xcalibur software (v.4.4, Thermo Fisher Scientific), and data were acquired using a data-dependent top-15 method with a full scan range of 300–1650 Th. MS1 survey scans were acquired at 60,000 resolution, with an automatic gain control (AGC) target of 3 × 10$^6$ charges and a maximum ion injection time of 25 ms. Selected precursor ions were isolated in a window of 1.4 Th and fragmented by higher-energy collisional dissociation (HCD) with normalized collision energies of 30. MS2 fragment scans were performed at 15,000 resolution, with an AGC target of 1 × 10$^5$ charges, a maximum injection time of 28 ms, and precursor dynamic exclusion for 30 s.

## Raw mass spectrometry data analysis

Mass spectrometry raw files were processed in MaxQuant Version 2.1.4.0[88,89], using the human SwissProt canonical and isoform protein database retrieved from UniProt (2022_09_26; www.uniprot.org). Label-free quantification was performed using the MaxLFQ algorithm[90]. Matching between runs was enabled to match between equivalent and adjacent peptide fractions within replicates. LFQ minimum ratio count was set to 1 and default parameters were used for all other settings. All downstream analyses were performed on the 'protein groups' file output from MaxQuant.

## Proteomic downstream data analysis

Differential enrichment analysis of proteomics data was done using the DEP package in R. Preprocessing and quality filtering were performed separately for SH-SY5Y cells and hiPSC-derived neurons. Proteins that were only identified by a modification site or matched the reversed part of the decoy database, as well as commonly occurring contaminants, were removed. Duplicate proteins were

removed based on the corresponding gene names by keeping those with the highest total MS/MS count across all samples. All following steps were done separately for each cell type (SH-SY5Y cells (Fig. 7a and Supplementary Fig. 8a–d) and hiPSC-derived neurons (Fig. 7b and Supplementary Fig. 8e–h) and for the pooled dataset (Fig. 7c and Supplementary Fig. 8i–l)). Low-quality entries were removed by keeping only those proteins that had valid MS/MS counts in all replicate samples of at least one experimental condition. Finally, only those proteins were kept that had a maximum of one missing LFQ value in at least one experimental condition. Filtered data were normalized using variance stabilizing transformation, and missing values were imputed using a manually defined left-shifted Gaussian distribution with a width of 0.3 and a left-shift of 2.2 SD. Batch effects were corrected using the method described by Johnson et al.[91]. Statistical testing for differential protein enrichment was done using protein-wise linear models and empirical Bayes statistics implemented in the limma package in R. Proteins were considered as differentially enriched with a FDR <0.05 and a $\log_2$ fold change >0.3. The biological information contained in differentially enriched proteins was summarized using Reactome pathway annotation in clusterProfiler[82]. Pathways were considered differentially expressed with a FDR <0.05.

## Nucleofection

sgRNAs against *NLRP5*, *RAB3C* and *RAB12* were purchased as multi-guide knockout kits (v2) from Synthego, diluted to $100\,\mu M$ stock concentrations and kept at −20 °C. Nucleofection was performed under RNAse-free conditions on a Lonza 4D-Nucleofector (Cat# AAF-1003X, AAF-1003B) according to the manufacturer's protocol. Briefly, SH-SY5Y cells were harvested, and $4 \times 10^5$ cells were resuspended in $5\,\mu L$ Nucleofector Solution. Then, 180 pmol sgRNAs were incubated with 20 pmol Cas9 protein in Nucleofector Solution to form ribonucleoprotein complexes (RNPs) according to the manufacturer's instructions. The cell solution was then incubated with the respective RNPs and transferred into a nucleofection strip (Cat# V4XC-2032). Strips were placed in the 4D-Nucleofector System, and nucleofection was done using the CA-137 program. Following nucleofection, pre-warmed medium was added after 10 min, and cells were plated. Compound treatment was started 48 h after nucleofection. Knockout efficiency of sgRNAs was assessed using the Synthego ICE Analysis online tool. Genomic DNA was extracted from nucleofected cells using the Quick-DNA Microprep Kit (Zymo Research, Cat# D3021) according to the manufacturer's instructions and amplified by PCR using the Platinum™ II Hot-Start PCR Master Mix (Thermo Fisher Scientific, Cat# 14000012). After a hot start, a denaturation temperature of 95 °C, an annealing temperature of 58 °C and an extension temperature of 72 °C were chosen and repeated for 40 cycles. For amplification the following primers were used, while for sequencing only the forward primer was used: *NLRP5* forward: CTTGAGAATTTGCTGCAAGATCCT, *NLRP5* reverse: CGATTCTTCCCTGTTCCCATGAG, *RAB3C* forward: CCACTCGCCTCCTGAGTGTCTG, *RAB3C* reverse: GAACAAGGCAGAAAGTTTCTCCC, *RAB12* forward: CTGTGCGCATGGGAGTGTTTTC, *RAB12* reverse: CTTACCCACGGTGGACTTGC.

## Statistical analyses

Statistical analysis of continuous variables was performed with R version 4.2.1 (2022-06-23) and Rstudio (version 2022.07.1; Rstudio, Inc.) using either mean and standard deviation (SD) or median and inter-quartile range (IQR), depending on the distribution of data tested by visualization with histograms, quantile-quantile plots and normality testing using the Shapiro-Wilk test. Sample sizes are indicated (*n*) for each analysis. The *t*-test (for normally distributed variables) and the Mann-Whitney U test (for non-parametric distributions) were performed to test for statistical differences.

## Reporting summary

Further information on research design is available in the Nature Portfolio Reporting Summary linked to this article.

## Data availability

RNA sequencing data are publicly available through the National Center for Biotechnology Information's Sequence Read Archive (SRA) accession number: PRJNA985061. Mass spectrometry proteomics data are deposited to the ProteomeXchange Consortium accession number: PXD042950 via the PRIDE partner repository. Source data are provided as a source data file and have been deposited in the figshare database: https://doi.org/10.6084/m9.figshare.23217491. Source images are available from the author upon request. All fibroblast and hiPSC lines generated in this study are available with a material transfer agreement. Source data are provided with this paper.

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

## Acknowledgements

The authors thank the patients and their families for participating in this study. The authors thank Selva G. Nataraja, PhD, and the teams at Mitobridge Inc. and Astellas Inc. for feedback and for providing small molecule libraries. The authors thank Jen Smith, PhD, Clarence Yapp, PhD, and the team from the ICCB-Longwood Screening Facility for help with designing and conducting screening experiments, Igor Paron, PhD and Tim Heymann, PhD from the Max Planck Institute of Biochemistry for their technical support for mass spectrometry, and the BCH-Astellas Joint Steering Committee members, Thomas Schwarz, PhD, Larry Benowitz, PhD, and Zhigang He, PhD, for critical feedback and guidance. This study was supported by research grants from the CureAP4 Foundation (to D.E.-F.), the Spastic Paraplegia Foundation (to D.E.-F.), the Tom-Wahlig Foundation (to A.S. and D.E.-F.), the Manton Center for Orphan Disease Research (to D.E.-F.), the BCH Office of Faculty Development (to D.E.-F.), the BCH Translational Research Program (to D.E.-F.), the National Institute of Neurological Disorders and Stroke grant (1K08NS123552-01 to D.E.-F.), and a joint research agreement with Mitobridge Inc. and Astellas Pharmaceuticals Inc. (to D.E.-F. and M.Sa.). A.S. was funded by the Deutsche Forschungsgemeinschaft (DFG, German Research Foundation)—448402208. Further support is acknowledged from the German National Academic Foundation (to B.B., C.B., J.E.A., M.Z., M.Sch.), the Carl-Duisburg Program of the Bayer Foundation (to B.B., M.Z.), the German National Exchange Service (to C.B., J.E.A., M.Z.), the European Union's Horizon 2020 research and innovation program under the Marie Sklodowska-Curie grant agreement no. 896725 (to A.K.D.), and the Rosamund Stone Zander chair (to M.Sa.). The IDDRC at Boston Children's Hospital is supported by National Institutes of Health grant 1U54HD090255.

## Author contributions

A.S., A.K.D., M.Sa., and D.E.F. conceptualized and designed the experiments. A.S., B.B., A.K.D., C.B., W.A.S., H.J., D.Wh., D.Wo., L.W., J.E.A., M.Z., and K.W. performed experiments. J.H. designed and supervised the re-synthesis of compound BCH-HSP-C01. E.D.B. and L.B. provided technical assistance and analysis tools. A.S., B.B., and D.E.F. wrote the first draft of the manuscript. A.S., B.B., C.B., W.A.S., H.J., D.Wh., D.Wo., L.W., J.E.A., M.Z., M.Sc., K.W., J.H., E.D.B., L.B., G.H.H.B., A.K.D., D.E.F., and M.Sa. contributed to the final draft of the manuscript. D.E.F. wrote the grants that supported this project. M.Sa. and D.E.F. supervised the project.

## Competing interests

This work was supported by a joint research agreement between Boston Children's Hospital and Mitobridge Inc., now owned by Astellas Pharmaceuticals Inc. D.E.F. has served as a consultant to Health Advances LLC, has received speaker honoraria from the Movement Disorders Society and publishing royalties from Cambridge University Press. M.Sa. reports grant support from Novartis, Biogen, Astellas, Aeovian, Bridgebio, and Aucta unrelated to this project. He has served on Scientific Advisory Boards for Novartis, Roche, Regenxbio, SpringWorks Therapeutics, Jaguar Therapeutics, and Alkermes. The remaining authors declare no competing interests.
