## [Peer Review File · Nature Communications]

REVIEWER COMMENTS

Reviewer #1 (Remarks to the Author):

In the manuscript “High-Content Small Molecule Screen Identifies a Novel Compound That Restores AP-4-Dependent Protein Trafficking in Neuronal Models of AP-4-Associated Hereditary Spastic Paraplegia”, Saffari et al. developed a high-throughput screening assay to identify molecules that can correct protein trafficking abnormalities in adaptor protein complex 4 (AP-4) deficiency, a rare form of childhood-onset hereditary spastic paraplegia. They screened a library of small molecules and discovered a lead compound, C-01, which successfully restored the mislocalization of the autophagy protein ATG9A in various in vitro disease models. Through further analysis, they identified potential molecular targets and mechanisms of action for C-01, providing promising data for future studies and the potential development of a treatment for AP-4 deficiency.

Overall this is a carefully designed and interpreted valid study that I would recommend for publication in “Nature Communication” after some minor corrections.

Key comments:

1.) The key question of the results is whether or not the change in ATG9A ratio (Inside TGN / Outside TGN) by the lead compound C-01 has disease ameliorating effects. It is well known that lack of ATG9A at the distal axon leads to axonal degeneration. Compound C-01 leads to the release of ATG9A from the TGN. Whether this also correlates with a transport of ATG9A to the distal parts of a cell/axon remains unclear and should ideally be addressed in iPSC-derived neurons.

2.) It would be interesting to see if there is a similar molecular structure of the other identified compounds (at least for the 16 compounds validated in the counter screen), indicating a common mechanism of action for several compounds.

Comments related to specific paragraphs:

Introduction

The introduction is well written and nicely summarizes the state of scientific knowledge

Results

General comment: Data visualization and interpretation are valid and robust. Very good and detailed description of the data evaluation.

Primary screen

- How many wells per condition (compound) were analyzed?
- There is a rather large variability in cell counts (Fig. 1g, 200 - 600). Cell count does not match the initial plating number (2,000 cells per well) How can this be explained?
- ll 110 – 111: “LoF/LoF mean: 1.1 ± 0.02 SD, WT/LoF mean: 1.34 ± 0.05 SD” – data swapped

Counter-screen

- 34 compounds were found to have autofluorescence or imaging artifacts. Was this already analyzed / identified in the primary screen?

Secondary screen (SH-SY5Y)

General comment: Why not used as "counter-screen assay" as it seems that the separation between positive and negative control is much more robust? (perhaps due to the fact of comparing a full KO to WT instead of a het carrier to a hom patient)

- The inclusion of the DAGLB assay is an important and careful validation
- Why was the "multiparametric morphological profiling approach" only performed on 5 compounds? Would it be possible to implement this analysis as a screening tool for future projects, or at least for the counter-screen?
- It appears that compounds B-01 and G-01 also alter cellular morphology in a dose-dependent manner (Suppl. Fig. 4, I 184)

iPSC-derived neurons

- It would be helpful to include a staining (overview) to visualize neuronal morphology especially for those conditions / compounds that show toxic properties
- Would it be possible to quantify ATG9A levels in the distal part of the axon?

Target deconvolution

- RNA sequencing: It is unclear why SHSY-5Y cells were used instead of iPSC-derived neurons as compound C-01 showed only small effects on ATG9A ratios in previous analyses (Fig. 3d)
- 72h instead of 24h treatment (results of ATG9A ratios for SHSY-5Y cells are missing)
- Autophagic flux: Interestingly, there is no baseline difference between AP4B1-KO and control SH-SY5Y cells (Fig. 8 j-l)

Discussion:

- Expanding the paragraph describing the limitations of the primary screen (only one cell line, single readout, 24h treatment, one concentration) would be valuable
- The key question of whether a redistribution of ATG9A and DAGLB is disease-relevant with respect to the long distances in patient neurons (distal axons) needs to be discussed

Minor comments:

Figures:

- Fig. 5g: "Molecular Weight: 256.3 kDA" – cannot be correct, 256.3 DA (g/mol)
- Fig. 8 I: Blots for RAB3C and RAB12 are missing

Methods:

- II 547-548: "with or without hiPSC-derived astrocytes": Were astrocytes used? Not mentioned elsewhere.

Figure legends:

- I 849: "with hundreds of thousands to millions of cells per experiment" – statement not helpful

Reviewer #2 (Remarks to the Author):

This very large and detailed manuscript by Saffari et al uses a phenotypic screening approach to identify small molecule C01 as a promoter of appropriate ATG9A trafficking in cells harboring A4P mutations established to cause spastic paraplegia. Figures 1-5 document their high-throughput screen and the various secondary assays they use to prioritize C01 over their other hits. Figures 6-8 use transcriptomic and proteomic approaches to try to provide understanding of how C01 functions.

Key strengths:

The disease context is very interesting. Additionally, since many other neurodevelopmental disorders converge on dysfunctional protein trafficking, the screening approach and C01/other hits may have broader significance than the setting of spastic paraplegia.

The screening and probe development efforts in Figs 1-6 are generally rigorous and use a strong mix of biological contexts including differentiated neuron-like cells and, critically, neurons derived from iPSCs obtained from patients. The overall degree of difficulty in executing this type of high-content screening approach is high, and being able to spotlight a molecule that works consistently across these various contexts is an accomplishment.

C01 itself is a key strength because, in contrast to many academic studies, this molecule has quite strong physicochemical properties. C01 would be a strong starting point for CNS applications in the eyes of most big pharma med chemists, and these properties make this discovery more impactful by opening greater future drug discovery possibilities.

While some experiments described below may not be ideal imo, it's undeniable that all the data in the manuscript (and there are a lot) are presented clearly and that the utmost effort has been put forth to be transparent and rigorous about the data generated and presented (noting especially the supporting figures and data sets).

Perceived weaknesses:

The central weakness is that Figures 6-8 make little headway in understanding how C01 functions. Target identification for small molecules is always a challenge, and it's made substantially more challenging here due to the subtlety of the phenotypes observed and the uM potency of C01; the degree of difficulty arguably couldn't be higher. That said, the transcriptomic analyses labor against the initial conclusion that there isn't much of a transcriptomic signature for C01; ultimately, no clear hypothesis emerges and no validation experiments are performed. The proteomic analyses do identify changes after 72h of C01 treatment in RAB proteins that are associated with vesicular trafficking/autophagy. However it's not clear that these changes play a meaningful role in the phenotypes elicited by C01. C01 causes a large reduction in RAB12 but CRISPR KO of RAB12 doesn't significantly alter ATG9A ratio; conversely C01 lowers ATG9A ratio but causes only a small (significant) reduction in RAB3B levels. Cells lacking these RAB proteins remain as responsive to C01 as WT cells, strongly hinting that pathways/targets beyond these RAB proteins are likely dominant for C01's mechanism. This critique is not a request for additional data,

because the next steps toward target ID would be very challenging long term studies that are clearly beyond the scope of this story. However, I do think it's important to scale back text claims regarding establishing "molecular targets" (including in the abstract and last sentence of introduction) since no targets of C01 are delineated here.

For validated hits (maybe the top 5 or top 17), I think it's necessary to show structures to enable replication or extension by other researchers. Eg others may be interested in disruptors of the TGN for other biological reasons and could be interested in testing F01 and H01. Apologies if I overlooked these among the many supporting files.

I would recommend giving C01 a more formal or at least more descriptive name (eg HMS1234, or just a 6-digit number) because C01 just isn't descriptive enough. Future researchers may report on this cpd and/or future vendors may want to sell this cpd, and it just needs a less ambiguous name.

Reviewer #3 (Remarks to the Author):

Saffari et al. report the finding of a small molecule, C01, able to modulate the trafficking of ATG9A and DAGLB from the trans golgi network to other subcellular regions, as revealed by fluorescent imaging, in AP-4 deficient neurons from hereditary spastic paraplegia patients. Using transcriptomics and proteomics analysis, the authors were able to show that the RAB proteins, RAB12 and RAB3C play a role in the mechanism of action of C01.

Both, ATG9A and DAGLB are cargo proteins of the adaptor protein AP4. AP4 mediates intracellular membrane trafficking and mutations in the four AP4 subunits have been identified as causes of HPG with intellectual disability. Accumulation of ATG9A, an autophagy protein, in the TGN has been previously suggested as a potential cause of the neuropathogenesis of AP4 deficiency. To date, there is a high unmet medical need for HPG. Therefore, finding novel molecules that can progress to the clinic or insights into the disease would be of high significance to the field.

The paper by Saffari et al. is well written and the figures are rich in information but clear. The logic of the study is also clear. Most of the conclusions are overall well supported by the data. The authors use a challenging phenotypic assay to identify LMW compounds that are able to correct the phenotype of ATG9A accumulation in TGN. From the primary hits obtained, most molecules induce toxicity or do not reproduce in other cellular models and only C01 meets the criteria for MoA studies, which the authors do very comprehensively.

However, I believe that in the current version, the work has important limitations that lower its impact in the field:

- The effect of C01 is also present in AP-4 WT cells indicating that the MoA of C01 is independent of the disease genotype. It would help to know if unrelated targets of AP-4 protein are also redistributed after C01 treatment, as to have an indication of whether this pathway would offer a therapeutic window to be explored further in the disease setup. Similarly, the effect of C01 is the mislocalization of AMPA

receptors, also reported to be involved in the neuropathology of the disease, would be of value

- Proteomics and transcriptomics analysis could focus on the differences across WT and KO AP4 models, to investigate potential disease targets independently of a C01 effect
- The molecular target of C01 remains unknown. RAB12 and RAB3C knockouts increase the effect of C01, which devalidate these proteins as the main targets leading to the ATG9A redistribution phenotype
- C01 is a small molecule poorly characterized for in vivo studies and with a weak potency in cells. In the paper there is not an analysis of the SAR in the library tested or exploration of the chemical space that supports this chemical scaffold. Therefore, the statements made by the authors in the Abstract, Introduction and Discussion section about the potential of C01 as IND and therapeutic for AP4-HSP are overstated

Minor comments:

- How many fibroblast donors were tested and how strong is the variation of the phenotype
- Unclear why the 61 hits showing toxicity in the primary screen were not removed from the primary selection
- Figure 3b: the assay window looks better in the neuronal model, why were fibroblast used instead for the primary screen?
- The data showed in Figure 3k: there is a great variability and no dose response. I would not conclude the effect of B-01 is also observed for DAGLB

Reviewer #4 (Remarks to the Author):

Thank you for the opportunity to review this elegant manuscript. I congratulate the authors on their approach to identifying therapeutic candidates for rare neurological diseases, the ultimate outcome of which likely will be therapeutics for rare diseases that have unanticipated spillover efficacy in more common neurological diseases. I have a few suggestions about the biology and the chemistry presented in this paper.

I appreciate that the authors view C-01 as a tool compound to launch a future medicinal chemistry campaign; however, I think the paper is greatly diminished by no in vivo data. C-01 has properties favorable for brain penetration. What peripheral dose achieves 5 uM (the EC50 in neuron derived iPSCs) in mouse brain extracellular fluid? What are the corresponding microscopic, transcriptomic, and proteomic changes? I think results from experiments like these will be necessary for readers to judge the translational potential of the compelling cell culture results in the present version.

It appears that Astellas gave the investigators the library but did not disclose structures until after screening was completed and only for the hit(s). It would be very helpful to reveal more about how the set was assembled. Astellas should be able to provide more information on what was considered, e.g., Tanimoto scores, MW, Lipinski, etc.

Why did the authors keep the 61 compounds that reduced cell counts? Shouldn't these be included with the "toxic" 1435 compounds that were excluded?

Synthesis (line 960) of C-01 not shown in Figure 5g.

Point by point response - Manuscript ID: NCOMMS-23-23211

We thank the editors and reviewers for the time and expertise invested in evaluating our work. We are pleased that the manuscript was well received and deemed interesting. Please find a detailed point-by-point response attached below.

Reviewer #1:

In the manuscript “High-Content Small Molecule Screen Identifies a Novel Compound That Restores AP-4-Dependent Protein Trafficking in Neuronal Models of AP-4-Associated Hereditary Spastic Paraplegia”, Saffari et al. developed a high-throughput screening assay to identify molecules that can correct protein trafficking abnormalities in adaptor protein complex 4 (AP-4) deficiency, a rare form of childhood-onset hereditary spastic paraplegia. They screened a library of small molecules and discovered a lead compound, C-01, which successfully restored the mislocalization of the autophagy protein ATG9A in various in vitro disease models. Through further analysis, they identified potential molecular targets and mechanisms of action for C-01, providing promising data for future studies and the potential development of a treatment for AP-4 deficiency.

Overall this is a carefully designed and interpreted valid study that I would recommend for publication in “Nature Communication” after some minor corrections.

Response: We thank the reviewer for their thoughtful comments. We are pleased to hear that our report was received well and deemed important for the field.

Key comments:

1.) The key question of the results is whether or not the change in ATG9A ratio (Inside TGN / Outside TGN) by the lead compound C-01 has disease ameliorating effects. It is well known that lack of ATG9A at the distal axon leads to axonal degeneration. Compound C-01 leads to the release of ATG9A from the TGN. Whether this also correlates with a transport of ATG9A to the distal parts of a cell/axon remains unclear and should ideally be addressed in iPSC-derived neurons.

Response: We thank the reviewer for this insightful comment. At their suggestion, we have completed additional analyses in iPSC-derived neurons from two individuals with AP-4-related hereditary spastic paraplegia and controls. We quantified the number of ATG9A puncta per neurite length using an automated image analysis pipeline in a high-throughput format. We find that AP-4 deficient neurons show a reduced number of ATG9A puncta compared to controls. Treatment with BCH-HSP-C01 increased the number of ATG9A puncta in neurites to levels similar to controls. These important findings and the complete dataset have been added to the revised version of the paper. Please refer to the revised

Figure 5 (panels k-m), the revised version of Supplementary File 7, as well as the following paragraph in main text:

“Prior work in neurons isolated from AP-4-deficient mice^{13,14} highlighted depletion of axonal ATG9A pools. In hiPSC-derived neurons from individuals with SPG50 and SPG47, we observed a reduction of ATG9A puncta density in neurites. BCH-HSP-C01 treatment for both 24h and 72h restored neurite ATG9A puncta density to levels similar to control (Fig. 5 k-m, Supplementary File 7).” (lines 219-222)

2.) It would be interesting to see if there is a similar molecular structure of the other identified compounds (at least for the 16 compounds validated in the counter screen), indicating a common mechanism of action for several compounds.

Response: We thank the reviewer for this comment and agree. In line with the response to Reviewer 2’s comment, we have now added the chemical structures of all compounds tested in iPSC-derived neurons (BCH-HSP-B01, BCH-HSP-C01, BCH-HSP-F01, BCH-HSP-G01, BCH-HSP-H01). Please refer to the revised version of Figure 5 (Panels d-f) and new Supplementary Figure 4 for details.

Comments related to specific paragraphs:

Introduction

The introduction is well written and nicely summarizes the state of scientific knowledge.

Response: We thank the reviewer for this comment.

Results

General comment: Data visualization and interpretation are valid and robust. Very good and detailed description of the data evaluation.

Response: We thank the reviewer for this positive feedback.

Primary screen

- How many wells per condition (compound) were analyzed?

Response: We thank the reviewer for raising this point. In the primary screen, 28,864 compounds were arrayed to single wells and one well per compound was screened. To account for the absence of replicates, recommended metrics for screens without replicates were used (Zhang XD. Illustration of SSMD, z score, SSMD*, z* score, and t statistic for hit selection in RNAi high-throughput screens. J Biomol Screen. 2011 Aug;16(7):775-85. PMID: 21515799). We acknowledge that using replicates could have potentially decreased the rate of false negatives. However, we compromised for cost and efficiency. With respect to false positives, these were eliminated in the counter screen which was performed in biological duplicate and using dose-response titrations.

To address this limitation in the manuscript, we added the following paragraphs:

“A diversity library of 28,864 novel small molecules was provided by Astellas Pharma Inc.. Compounds were arrayed to single wells in 384-well microplates and one well per compound was screened.” (lines 63-64)

“First, in the primary screen compounds were arrayed to single wells and only one well per compound was screened. We recognize that using multiple replicates as well as multiple concentrations and treatment durations could have potentially decreased the rate of false negative results. However, we prioritized efficiency and compounds that would show a robust impact at a single low-micromolar concentration. With respect to false positive results, these were eliminated in the counter screen, which was performed with biological duplicates and using dose-response titrations covering a broad range of concentrations.” (lines 426-432)

- There is a rather large variability in cell counts (Fig. 1g, 200 - 600). Cell count does not match the initial plating number (2,000 cells per well) How can this be explained?

Response: We thank the reviewer for this question and are happy to clarify: Although 2,000 cells per well were seeded, only 4 fields of view per well were imaged in the primary screen (see Methods section, line 578). The number of cells assessed per plate is representative of the fraction of the cells covered by 2x2 images taken in the center of each well. Cell counts thus may vary depending on the distribution of cells at the center of the well.

- ll 110 – 111: “LoF/LoF mean: 1.1 ± 0.02 SD, WT/LoF mean: 1.34 ± 0.05 SD” – data swapped.

Response: We thank the reviewer for catching this mistake and have corrected this sentence.

Counter-screen

- 34 compounds were found to have autofluorescence or imaging artifacts. Was this already analyzed / identified in the primary screen?

Response: We appreciate the reviewer’s comment. Additional analyses for autofluorescence and artifacts were done by manual inspection of the images. Since the number of compounds in the primary screen was too high for this manual approach, it was only done as part of the counter screens. That said, many wells with imaging artifacts were likely excluded at the primary screening stage, given that artifacts (i.e., out of focus images) usually lead to a significant change in cell counts.

Secondary screen (SH-SY5Y)

General comment: Why not used as "counter-screen assay" as it seems that the separation between positive and negative control is much more robust? (perhaps due to the fact of comparing a full KO to WT instead of a het carrier to a hom patient).

Response: We understand the reviewer's point and agree that the separation of positive and negative controls is even better in SH-SY5Y cells, which is likely due to the homogeneity of this cell line, but also appears to be a feature of other neuronal lines, such as iPSC-derived neurons, perhaps reflecting the increased sensitivity of neurons to loss of AP-4. The screening approach was designed to use predominantly patient-derived cells. In the primary and secondary screen fibroblasts from an AP-4 patient, which are expected to cause a full loss-of-function, were compared to heterozygous controls, which are expected to behave like wildtypes, served as an easily obtainable and patient-relevant cell model (for a detailed assessment of the fibroblast lines compare Behne *et al.*, Hum Mol Genet. 2020, doi: 10.1093/hmg/ddz310). *AP4B1* knockout SH-SY5Y cells, which are a more neuron-like but arguably less patient-relevant cell type, due to their homogeneity, were used in orthogonal assays and for exploration of potential mechanisms. We believe this approach and combination of multiple *in vitro* disease models provides key advantages over screens done in a single cell line of any type.

- The inclusion of the DAGLB assay is an important and careful validation.

Response: We thank the reviewer for this comment and agree that evaluating the translocation of a second AP-4 cargo protein adds an important validation step.

- Why was the "multiparametric morphological profiling approach" only performed on 5 compounds? Would it be possible to implement this analysis as a screening tool for future projects, or at least for the counter-screen?

Response: We thank the reviewer for pointing out the value of multiparametric analyses in cell-based assays. Since our goal was to identify compounds that re-distribute ATG9A, while maintaining acceptable cell counts, these two metrics were deemed sufficient as readouts to efficiently identify compounds of interest in the primary and counter-screens. The value of morphologic profiling was highest at the stage of having identified 5 lead compounds that all met predefined criteria. Here, a multiparametric approach helped us profile the impact on multiple cellular phenotypes/structures, beyond our primary readouts, and thus helped us triage compounds with potential off-target effects. For future projects of similar scope, multiparametric morphologic profiling can be implemented at any stage, including primary and counter screens, depending on the research question. To facilitate using this approach for future studies, we added a detailed description on how to implement morphologic profiles to the methods section:

"The multiparametric morphological profiling strategy employed in this study was adapted from previously published protocols³⁴. Single cell measurements of ninety distinct descriptors of shape and intensity for the nucleus (DAPI), the cytoskeleton

and global cell morphology (anti-beta-Tubulin III), the TGN (anti-TGN46), and ATG9A vesicles (anti-ATG9A) were acquired and automatically extracted. Single cell data were summarized by computing per-image medians for each variable (Supplementary File 6). Next, a correlation matrix was generated using the Pearson correlation coefficient with complete pairwise observations. Variables with zero variance and observations with missing values were removed. Variables were then transformed to have a mean of zero and a standard deviation of one. Principal component analysis (PCA) was conducted to reduce dimensionality and cluster data based on their properties. To identify the contribution of individual features to the variance within the dataset, correlation analysis was performed between the first principal component, accounting for the majority of the variance within the dataset, and all extracted features. Features displaying a correlation coefficient > 0.75 were selected to define morphological profiles. Profiles were summarized using heatmap visualization..” (lines 615-628)

- It appears that compounds B-01 and G-01 also alter cellular morphology in a dose-dependent manner (Suppl. Fig. 4, I 184).

Response: We agree with the reviewer. We initially rated the changes on cellular morphology induced by BCH-HSP-B01 and BCH-HSP-G01 as minor compared to BCH-HSP-F01 and BCH-HSP-H01. Following the reviewer’s suggestion, we changed the wording to the following:

“BCH-HSP-C01 showed properties comparable to positive and negative controls, suggesting little off-target effects (Fig. 4b Supplementary Fig. 5c). BCH-HSP-B01, BCH-HSP-F01, BCH-HSP-G01 and BCH-HSP-H01, however, changed cellular morphology in a dose-dependent manner (Fig. 4b and Supplementary Fig. 5b,d,e,f), with changes mainly driven by the first principal component, accounting for 31.1% of the observed variance (Fig. 4c).” (lines 160-165)

iPSC-derived neurons

- It would be helpful to include a staining (overview) to visualize neuronal morphology especially for those conditions / compounds that show toxic properties.

Response: We thank the reviewer for this comment and have added low magnification images. Please refer to new Supplementary Figure 6.

- Would it be possible to quantify ATG9A levels in the distal part of the axon?

Response: We thank the reviewer for this question which aligns with the reviewer’s first key comment. In the experiment we describe above, we specifically assessed ATG9A localization in neurites. We are grateful for the reviewer’s suggestion to include this experiment, as it strengthens the manuscript.

Target deconvolution

- RNA sequencing: It is unclear why SHSY-5Y cells were used instead of iPSC-derived neurons as compound C-01 showed only small effects on ATG9A ratios in previous analyses (Fig. 3d).

Response: We understand the reviewer's point. SH-SY5Y cells were used for RNA sequencing due to their genetic homogeneity, eliminating potential biases that arise from different genetic backgrounds. Since iPSC neurons were derived from a patient vs. the sex-matched parents, we anticipated that the transcriptional differences between these two individuals would bias our results. With respect to the transcriptional changes with BCH-HSP-C01 in SH-SY5Y cells, please refer to the next comment.

- 72h instead of 24h treatment (results of ATG9A ratios for SHSY-5Y cells are missing).

Response: We thank the reviewer for raising this point. We performed the transcriptomics experiments after 72h of compound treatment since the effect of BCH-HSP-C01 was found to be dose- and time-dependent (possibly due to half-life of proteins, resulting in greater turnover with prolonged treatment). To clarify this point, we added two new datasets and figures:

- 1) A times series experiment with different concentrations of BCH-HSP-C01 showing that the maximum effect on ATG9A translocation is reached at 72h.**
- 2) A dose response curve of BCH-HSP-C01 after 72h treatment showing a maximum reduction of the ATG9A ratio of around 12 SD compared to the negative control.**

Please refer to revised Figure 5g&h and the following new text passage:

“To investigate the time- and dose-dependent effect of BCH-HSP-C01, we used *AP4B1^{KO}* SH-SY5Y cells to conduct time series experiments with different concentrations of BCH-HSP-C01 (Fig. 5 g,h, Supplementary File 6). All concentrations tested show a maximal effect on ATG9A translocation after 72-96h of treatment (Fig. 5g,h, Supplementary File 6), exceeding the effects seen after 24h (Fig. 3d).” (lines 201-205)

- Autophagic flux: Interestingly, there is no baseline difference between AP4B1-KO and control SH-SY5Y cells (Fig. 8 j-l)

Response: We agree with the reviewer that the lack of visible difference in autophagic flux at baseline seems unexpected at first. However, these findings by western blotting are consistent with our proteomics data from DMSO treated *AP4B1^{WT}* and *AP4B1^{KO}* SH-SY5Y cells that show no significant change in protein levels of LC3B. We acknowledge that this contrasts with previously published western blot and proteomics data from AP-4-KO HeLa cells (Davies, et al. Nat Commun. 2018 Sep 27;9(1):3958. PMID: 30262884; Mattera et al. Proc Natl Acad Sci U S A. 2017 PMID: 29180427). However, our findings are in line with other published data on LC3 levels in AP-4-KO neurons, where no changes in basal levels

of LC3B were detected (Fig. 7D-F, de Pace et al. PLoS Genet. 2018 PMID: 29698489). Thus, the impact of AP-4 deficiency on LC3 levels appears to be cell-type specific.

Discussion:

- Expanding the paragraph describing the limitations of the primary screen (only one cell line, single readout, 24h treatment, one concentration) would be valuable.

Response: We thank the reviewer for this suggestion and refer to the data we added in response to reviewer 1, comment 2. We now show that BCH-HSP-C01 not only reduces the ATG9A ratio at the TGN, but also increases the amount of ATG9A puncta in neurites, supporting a disease-relevant redistribution of AP-4 cargo proteins.

- The key question of whether a redistribution of ATG9A and DAGLB is disease-relevant with respect to the long distances in patient neurons (distal axons) needs to be discussed

Response: We thank the reviewer for this suggestion and refer to our response to reviewer 1, comment 2. At the suggestion of reviewer 1, we have completed additional analyses in iPSC-derived neurons from two individuals with AP-4-related hereditary spastic paraplegia and controls. We quantified the number of ATG9A puncta per neurite length using an automated image analysis pipeline in a high-throughput format. We find that AP-4 deficient neurons have a decreased number of ATG9A puncta compared to controls. Treatment with BCH-HSP-C01 significantly increased the number of ATG9A puncta in neurites to levels similar to controls. This important finding and dataset have been added. Please refer to the revised version of Figure 5 (Panels k-m), the revised version of Supplementary file 7 as well as the following paragraph that has been added to the main text:

“Prior work in neurons isolated from AP-4-deficient mice^{13,14} highlighted depletion of axonal ATG9A pools. In hiPSC-derived neurons from individuals with SPG50 and SPG47, we observed a reduction of ATG9A puncta density in neurites. BCH-HSP-C01 treatment for both 24h and 72h restored neurite ATG9A puncta density to levels similar to control (Fig. 5 k-m, Supplementary File 7).” (lines 219-222)

Minor comments:

Figures:

- Fig. 5g: “Molecular Weight: 256.3 kDA” – cannot be correct, 256.3 DA (g/mol)

Response: We thank the reviewer for catching this mistake. The information on compound structures and molecular weights of all five compounds tested in iPSC-derived neurons has been added to new Supplementary Fig. 4.

- Fig. 8 I: Blots for RAB3C and RAB12 are missing

Response: We thank the reviewer for this comment and have added panels for Rab12 and Rab3c to this figure. Please refer to the revised version of Fig. 8I and Supplementary Fig. 10d.

Methods:

- ll 547-548: “with or without hiPSC-derived astrocytes”: Were astrocytes used? Not mentioned elsewhere.

Response: We apologize for the confusion. For staining and imaging experiments, hiPSC-derived neurons were plated in co-cultures with astrocytes to ensure better neuronal health. For preparation of neuronal cell lysates for proteomics experiments, neurons were plated without astrocytes. To clarify, we adjusted the methods section to the following:

“On day 6, cells were dissociated with papain (Worthington, Cat# LK003178) and DNaseI (Worthington, Cat# LK003172) and replated on poly-D-lysine (0.5mg/ml; Sigma Aldrich, Cat#P6407) and laminin (5µg/ml; Thermo Fisher Scientific, Cat #23017-015) coated plates either in co-culture with hiPSC-derived astrocytes (Astro.4U, Ncardia) for immunocytochemistry experiments, or without astrocytes for preparation of cell lysates for proteomics experiments.” (lines 551-556)

Figure legends:

- l 849: “with hundreds of thousands to millions of cells per experiment” – statement not helpful

Response: We understand the reviewer’s point and have refined this statement. (line 873-874)

Reviewer #2 (Remarks to the Author):

This very large and detailed manuscript by Saffari et al uses a phenotypic screening approach to identify small molecule C01 as a promoter of appropriate ATG9A trafficking in cells harboring A4P mutations established to cause spastic paraplegia. Figures 1-5 document their high-throughput screen and the various secondary assays they use to prioritize C01 over their other hits. Figures 6-8 use transcriptomic and proteomic approaches to try to provide understanding of how C01 functions.

Key strengths:

The disease context is very interesting. Additionally, since many other neurodevelopmental disorders converge on dysfunctional protein trafficking, the screening approach and C01/other hits may have broader significance than the setting of spastic paraplegia.

The screening and probe development efforts in Figs 1-6 are generally rigorous and use a strong mix of biological contexts including differentiated neuron-like cells and, critically, neurons derived from iPSCs obtained from patients. The overall degree of difficulty in executing this type of high-content screening approach is high, and being able to spotlight a molecule that works consistently across these various contexts is an accomplishment.

C01 itself is a key strength because, in contrast to many academic studies, this molecule has quite strong physicochemical properties. C01 would be a strong starting point for CNS applications in the eyes of most big pharma med chemists, and these properties make this discovery more impactful by opening greater future drug discovery possibilities.

While some experiments described below may not be ideal imo, it's undeniable that all the data in the manuscript (and there are a lot) are presented clearly and that the utmost effort has been put forth to be transparent and rigorous about the data generated and presented (noting especially the supporting figures and data sets).

Response: We thank the reviewer for their positive assessment and for pointing out the strengths of our manuscript.

Perceived weaknesses:

The central weakness is that Figures 6-8 make little headway in understanding how C01 functions. Target identification for small molecules is always a challenge, and it's made substantially more challenging here due to the subtlety of the phenotypes observed and the uM potency of C01; the degree of difficulty arguably couldn't be higher. That said, the transcriptomic analyses labor against the initial conclusion that there isn't much of a transcriptomic signature for C01; ultimately, no clear hypothesis emerges and no validation experiments are performed. The proteomic analyses do identify changes after 72h of C01 treatment in RAB proteins that are associated with vesicular trafficking/autophagy. However it's not clear that these changes play a meaningful role in the phenotypes elicited by C01. C01 causes a large reduction in RAB12 but CRISPR KO of RAB12 doesn't significantly alter ATG9A ratio; conversely C01 lowers ATG9A ratio but causes only a small (significant) reduction in RAB3B levels. Cells lacking these RAB proteins remain as responsive to C01 as WT cells, strongly hinting that pathways/targets beyond these RAB proteins are likely dominant for C01's mechanism. This critique is not a request for additional data, because the next steps toward target ID would be very challenging long term studies that are clearly beyond the scope of this story. However, I do think it's important to scale back text claims regarding establishing "molecular targets" (including in the abstract and last sentence of introduction) since no targets of C01 are delineated here.

Response: We acknowledge the reviewer's point and have scaled back our wording in the respective paragraphs.

For validated hits (maybe the top 5 or top 17), I think it's necessary to show structures to enable replication or extension by other researchers. Eg others may be interested in disrupters of the

TGN for other biological reasons and could be interested in testing F01 and H01. Apologies if I overlooked these among the many supporting files.

Response: We thank the reviewer for this comment and refer to the response to reviewer 1. We have now added the chemical structures of all compounds tested in iPSC-derived neurons (BCH-HSP-B01, BCH-HSP-C01, BCH-HSP-F01, BCH-HSP-G01, BCH-HSP-H01). Please refer to the revised version of Figure 5 (panels d-f) and new Supplementary Fig. 4 for details.

I would recommend giving C01 a more formal or at least more descriptive name (eg HMS1234, or just a 6-digit number) because C01 just isn't descriptive enough. Future researchers may report on this cpd and/or future vendors may want to sell this cpd, and it just needs a less ambiguous name.

Response: We thank the reviewer for this suggestion and agree. We have revised the naming system for all compounds by adding a more descriptive suffix. Compound C01 has been renamed to BCH-HSP-C01.

Reviewer #3 (Remarks to the Author):

Saffari et al. report the finding of a small molecule, C01, able to modulate the trafficking of ATG9A and DAGLB from the trans golgi network to other subcellular regions, as revealed by fluorescent imaging, in AP-4 deficient neurons from hereditary spastic paraplegia patients. Using transcriptomics and proteomics analysis, the authors were able to show that the RAB proteins, RAB12 and RAB3C play a role in the mechanism of action of C01.

Both, ATG9A and DAGLB are cargo proteins of the adaptor protein AP4. AP4 mediates intracellular membrane trafficking and mutations in the four AP4 subunits have been identified as causes of HPG with intellectual disability. Accumulation of ATG9A, an autophagy protein, in the TGN has been previously suggested as a potential cause of the neuropathogenesis of AP4 deficiency. To date, there is a high unmet medical need for HPG. Therefore, finding novel molecules that can progress to the clinic or insights into the disease would be of high significance to the field.

The paper by Saffari et al. is well written and the figures are rich in information but clear. The logic of the study is also clear. Most of the conclusions are overall well supported by the data. The authors use a challenging phenotypic assay to identify LMW compounds that are able to correct the phenotype of ATG9A accumulation in TGN. From the primary hits obtained, most molecules induce toxicity or do not reproduce in other cellular models and only C01 meets the criteria for MoA studies, which the authors do very comprehensively.

Response: We thank the reviewer for their positive feedback and for acknowledging our efforts.

However, I believe that in the current version, the work has important limitations that lower its impact in the field:

- The effect of C01 is also present in AP-4 WT cells indicating that the MoA of C01 is independent of the disease genotype. It would help to know if unrelated targets of AP-4 protein are also redistributed after C01 treatment, as to have an indication of whether this pathway would offer a therapeutic window to be explored further in the disease setup. Similarly, the effect of C01 is the mislocalization of AMPA receptors, also reported to be involved in the neuropathology of the disease, would be of value

Response: We thank the reviewer for raising this important point. All targets of BCH-HSP-C01 in AP-4 WT and AP-4 KO cells are illustrated in Supplementary Fig. 7 (untargeted transcriptomics) and Supplementary Fig. 8 (untargeted proteomics), as well as the corresponding Supplementary Files 8 (transcriptomics) and 10 (proteomics). These also include BCH-HSP-C01 targets unrelated to AP-4 biology.

In addition to ATG9A, we have verified DAGLB as a second AP-4 cargo protein. AMPAR subunits (GRIA1-4) were, despite the high number of unique proteins identified in our proteomics experiment, not sufficiently quantified in SH-SY5Y cells. In iPSC neurons, only subunits GRIA2 and GRIA4 were adequately quantified (see attached Figure). Indeed, GRIA2 showed a trend of being upregulated in patient cells, however, without reaching significance. BCH-HSP-C01 treatment did not lead to any changes in expression levels. Due to these very subtle changes without clear statistical interpretation, we prefer to not report these changes in AMPA receptor subunits. The complete dataset is provided as part of the supplementary data, however, and can be further interrogated by any researchers interested in specific putative AP-4 cargo proteins.

- Proteomics and transcriptomics analysis could focus on the differences across WT and KO AP4 models, to investigate potential disease targets independently of a C01 effect

Response: We thank the reviewer for raising this point and have expanded on the differences between WT and KO models. In the transcriptomics experiment only the small nucleolar RNA gene RNU6-1 was identified to be differentially expressed. We have labeled this protein in the revised volcano plot in Supplementary Fig. 7a.

To expand on the differences between WT and KO models on the proteome level, we have added lists of all differentially enriched proteins in SH-SY5Y cells and iPSC neurons, along with descriptions of the protein function, and highlighted potential target pathways independent of BCH-HSP-C01 by performing gene ontology analysis. Please refer to revised Supplementary Fig. 8, Panels m&n, revised Supplementary File 10 and the following updated sections in the main text:

“[in SH-SY5Y cells], additional dysregulation of proteins involved in autophagy, Golgi dynamics and vesicular transport was identified (Supplementary Fig. 8a,m, Supplementary File 10). Of note, we observed upregulation of ATG2A, which has recently been shown to form a complex with ATG9A that facilitates lipid transfer from the endoplasmic reticulum (ER) to the growing phagophore membrane^{42, 43, 44}. This further supports that autophagosome biogenesis is dysregulated in AP-4-deficient cells.” (lines 275-279)

“[in hiPSC neurons] cell lines were a stronger discriminator, likely due to heterogeneity of the positive and negative controls, as expected” (lines 286-287)

- The molecular target of C01 remains unknown. RAB12 and RAB3C knockouts increase the effect of C01, which devalidate these proteins as the main targets leading to the ATG9A redistribution phenotype

Response: We agree with the reviewer that the exact molecular targets of BCH-HSP-C01 remain unknown. As stated by reviewer 2, a comprehensive analysis of putative molecular targets would extend beyond the scope of this manuscript. We acknowledge that our data hint at pathways/targets beyond RAB12 and RAB3C, however, our data also show that these proteins are possibly involved in the mechanism of this novel compound, in the sense that their knockdown/downregulation enhances BCH-HSP-C01's effect. We have adapted our discussion of these findings: “RAB3C and RAB12 showed the strongest and most consistent association with BCH-HSP-C01 treatment in both SH-SY5Y cells and hiPSC-derived neurons, and our analyses suggest that these two proteins likely contribute to BCH-HSP-C01-mediated redistribution of ATG9A from the TGN and increase of autophagic flux.” (lines 398-401)

- C01 is a small molecule poorly characterized for in vivo studies and with a weak potency in cells. In the paper there is not an analysis of the SAR in the library tested or exploration of the chemical space that supports this chemical scaffold. Therefore, the statements made by the authors in the Abstract, Introduction and Discussion section about the potential of C01 as IND and therapeutic for AP4-HSP are overstated

Response: We understand the reviewer's point and have removed all statements about IND applications.

Minor comments:

- How many fibroblast donors were tested and how strong is the variation of the phenotype

Response: We thank the reviewer for this comment. The primary screen was done in one well-characterized fibroblast line from a patient with SPG50 and the sex-matched control. This fibroblast line was previously published by our group along with 17 additional fibroblast lines from patients with SPG47, SPG50 and SPG52 showing little inter-individual variability in ATG9A mislocalization phenotype (Table 1 of Ebrahimi-Fakhari et al., *Brain Communications*, 2021, PMID: 34729478). For clarification, we added this reference.

- Unclear why the 61 hits showing toxicity in the primary screen were not removed from the primary selection

Response: We thank the reviewer for raising this point. Since in the primary screen all compounds were tested in one well at a concentration of 10 μ M, we considered the possibility that active AND toxic compounds might, in some cases, still show activity at lower concentrations without exerting toxicity. These were thus retested in the secondary screen in dose-response experiments covering a broad range from 40nM – 40 μ M.

- Figure 3b: the assay window looks better in the neuronal model, why were fibroblast used instead for the primary screen?

Response: We appreciate the reviewer's point and agree that the assay metrics are even better in SH-SY5Y cells. We believe that this might be due to the homogeneity of this stable cell line and the fact that neuronal cells seem to have a more severe phenotype than non-neuronal cells. Since the aim was to screen for compounds that were active in patient-derived cells, we chose primary fibroblasts as a cost-effective and robust disease model (extensively characterized by four independent groups, including ours (Ebrahimi-Fakhari et al., *Brain Communications*, 2021, PMID: 34729478)). We acknowledge that iPSC-derived neurons from patients are a disease model with greater similarity to relevant target tissues in patients. However, conducting a primary screen of this scale in iPSC-derived neurons would add several layers of complexity that might reduce our ability to interpret results of the primary screen. Thus, a scaled approach moving from a simple, robust phenotype present across all model systems to more sophisticated re-screen and orthogonal assays in complex cell types seemed to have several advantages.

- The data showed in Figure 3k: there is a great variability and no dose response. I would not conclude the effect of B-01 is also observed for DAGLB

Response: We agree with the reviewer and have changed the corresponding passage to: “All active compounds, except for BCH-HSP-B01, showed activity in the DAGLB assay ...” (line 148)

Reviewer #4 (Remarks to the Author):

Thank you for the opportunity to review this elegant manuscript. I congratulate the authors on their approach to identifying therapeutic candidates for rare neurological diseases, the ultimate outcome of which likely will be therapeutics for rare diseases that have unanticipated spillover efficacy in more common neurological diseases. I have a few suggestions about the biology and the chemistry presented in this paper.

Response: We thank the reviewer for their kind and positive feedback.

I appreciate that the authors view C-01 as a tool compound to launch a future medicinal chemistry campaign; however, I think the paper is greatly diminished by no *in vivo* data. C-01 has properties favorable for brain penetration. What peripheral dose achieves 5 μ M (the EC50 in neuron derived iPSCs) in mouse brain extracellular fluid? What are the corresponding microscopic, transcriptomic, and proteomic changes? I think results from experiments like these will be necessary for readers to judge the translational potential of the compelling cell culture results in the present version.

Response: We thank the reviewer for this comment and agree that *in vivo* proof-of-concept data will be essential to assess the translational potential of BCH-HSP-C01. However, the suggested experiments extend beyond the scope of this manuscript and will be addressed in future studies. We acknowledge this in the revised discussion section: “In conclusion, our findings provide a solid foundation for lead optimization of BCH-HSP-C01 and its development as a potential therapeutic, with the next step being *in vivo* proof-of-concept experiments.” (lines 455-457)

It appears that Astellas gave the investigators the library but did not disclose structures until after screening was completed and only for the hit(s). It would be very helpful to reveal more about how the set was assembled. Astellas should be able to provide more information on what was considered, e.g., Tanimoto scores, MW, Lipinski, etc.

Response: We thank the reviewer for this important remark. Astellas agreed to share the chemical structures and information of all five compounds tested in iPSC-derived neurons (compare revised Fig. 5, panels d-f and new Supplementary Fig. 4).

Why did the authors keep the 61 compounds that reduced cell counts? Shouldn't these be included with the "toxic" 1435 compounds that were excluded?

Response: We thank the reviewer for raising this point, which was also raised by Reviewer 3. Since in the primary screen all compounds were tested in one well at a concentration of 10 μ M, we considered the possibility that active AND toxic compounds might in some cases still show activity in lower concentrations without exerting toxicity and were thus retested in the secondary screen in dose-response experiments covering a broad range from 40nM – 40 μ M.

Synthesis (line 960) of C-01 not shown in Figure 5g.

Response: We apologize for this mistake. We have removed this sentence from the text.

REVIEWERS' COMMENTS

Reviewer #1 (Remarks to the Author):

I congratulate the authors on their comprehensive revision of this manuscript. The authors have addressed all my concerns, including the important experiments on the redistribution of ATG9A in iPSC-derived neurons upon treatment with BCH-HSP-C01.

Reviewer #2 (Remarks to the Author):

The revision has fully addressed my concerns. Certain data additions, particularly in response to Reviewer 1's critique, appear to significantly strengthen this revision.

Reviewer #3 (Remarks to the Author):

I would like to thank the authors for addressing all comments and the detailed point-by-point response. I believe the manuscript is now ready for acceptance.

Reviewer #4 (Remarks to the Author):

The authors have substantially improved an already excellent manuscript. I have no further suggestions.